# Methionine metabolism influences genomic architecture and gene expression through H3K4me3 peak width

Ziwei Dai[1], Samantha J. Mentch[1], Xia Gao[1], Sailendra N. Nichenametla[2] & Jason W. Locasale [1]

Nutrition and metabolism are known to influence chromatin biology and epigenetics through post-translational modifications, yet how this interaction influences genomic architecture and connects to gene expression is unknown. Here we consider, as a model, the metabolically-driven dynamics of H3K4me3, a histone methylation mark that is known to encode information about active transcription, cell identity, and tumor suppression. We analyze the genome-wide changes in H3K4me3 and gene expression in response to alterations in methionine availability in both normal mouse physiology and human cancer cells. Surprisingly, we find that the location of H3K4me3 peaks is largely preserved under methionine restriction, while the response of H3K4me3 peak width encodes almost all aspects of H3K4me3 biology including changes in expression levels, and the presence of cell identity and cancer-associated genes. These findings may reveal general principles for how nutrient availability modulates specific aspects of chromatin dynamics to mediate biological function.

[1] Department of Pharmacology and Cancer Biology, Duke Molecular Physiology Institute, Duke Cancer Institute, Duke University School of Medicine, Durham, NC 27710, USA. [2] Orentreich Foundation for the Advancement of Science, Cold Spring, NY 10516, USA. Correspondence and requests for materials should be addressed to J.W.L. (email: jason.locasale@duke.edu)

Genes interact with environmental factors such as nutrition to shape the epigenome that together influences gene activity and organismal physiology. Metabolism is also shaped by genes and environment and has a substantial contribution to epigenetics[1–4]. This nexus is essential in numerous biological contexts, including maintaining different stages of pluripotency[5–8], mediating an immune response[9,10], promoting or suppressing cancer progression[11–16], and transducing information about metabolic health and longevity from parent to offspring[17–19]. The molecular foundation of this interaction is in large part determined by the modifications on chromatin.

Chromatin is affected by metabolism through changes in the concentrations of metabolites that serve as substrates and cofactors for post-translational modifications. These concentrations are dynamic and are mediated by changes in metabolic pathway activity or flux that arise from transcriptional programs and nutrient availability. For example, histone methylation requires S-adenosylmethionine (SAM) as the universal methyl donor. SAM is derived from methionine[20] and its concentration can fluctuate in physiological conditions around values that can limit the activity of histone methyltransferases[21].

In plasma, methionine is in some reports the most dynamic of the 20 amino acids and the variation can to a large extent be explained by diet[22]. Recently work from us and others has shown that dietary modulation of methionine concentrations that approach the lower end of what can be observed in humans leads to bulk changes in the levels of histone methylation[22,23]. Other studies have reported similar findings in that changes to SAM levels or to the levels of alpha-ketoglutarate that modify the activity of demethylase enzymes induce global changes in the levels of histone modifications[5,12,24–32]. When these modifications are known to mark key aspects of chromatin status, global changes could have broad consequences to epigenomic programs. How these bulk changes to the levels of post-translational modifications on chromatin alter the genomic architecture of histone marks and relate to gene expression is, however, largely unknown.

One attractive model to investigate this interaction at the genome scale is the tri-methylation of histone H3 on lysine 4 (H3K4me3). The global (i.e., bulk) levels of this mark are dynamically and reversibly responsive to the levels of methionine[22]. In addition, there are numerous lines of evidence indicating that the structural features of H3K4me3, such as the width or breadth of the peak as deposited over a genic region, encode information such as gene activity, and gene function such as the presence of a developmental program, cell type identity, or a tumor suppressor[33–37]. Thus, changes in H3K4me3 may be relevant to developmental transitions and tumor suppression. How metabolic dynamics that occur due to differences in nutritional status or metabolic pathway activity might affect these programs and gene activity related to H3K4me3 is largely unknown.

We have shown previously that methionine availability modulates bulk levels of H3K4me3 by modifying SAM concentrations[22]. In this present study, we question whether changes in methionine availability that are known to affect global levels of H3K4me3 affect specific aspects of the genomic architecture and gene expression regulation. We consider a mouse model of dietary methionine restriction (MR) and focused our analysis on liver. In this organ, this diet results in changes to bulk levels of H3K4me3. Similar changes occur in cultured human cancer cells (HCT116) subjected to acute MR in culture media, that together provide a complementary set of two species, environmental conditions, biological statuses (health and cancer), models (in vitro and in vivo) and two tissues. We study genome-wide H3K4me3 dynamics using a quantitative ChIP-seq analysis that considers peak geometry and characterize the connection to gene expression dynamics. We find that height and area of the peaks are overall reduced, which account for most of the global changes. Strikingly, however, while the most conserved feature of H3K4me3 dynamics is the peak width, changes in peak width but not other features of peak geometry reflect important cellular processes previously linked to H3K4me3, including cell identity-related gene expression programs and the dynamics of gene expression.

## Results

### MR reduces H3K4me3 but maintains its genomic distribution.
To begin to study the impact of methionine availability on the genomic architecture of H3K4me3, we applied chromatin immunoprecipitation with sequencing (ChIP-seq) to map genomic locations enriched with H3K4me3 in HCT116 cells cultured under high (100 μM) and low, MR (3 μM) methionine availability which is known to lead to several-fold global changes in the bulk levels of H3K4me3[22]. Aligning the reads to a reference genome followed by peak calling identified a set of H3K4me3 peaks, i.e., genomic regions significantly enriched (MACS2 enrichment Q-value <1e-5) with ChIP-seq reads in comparison to the control (Fig. 1a). Comparing total peak number (coefficient of variation (CV) = 0.01, Fig. 1b), genomic location (Jaccard index = 0.88, Fig. 1c), and the set of genes marked by peaks (Jaccard index = 0.92, Supplementary Fig. 1a) between high and low methionine conditions showed that the distribution of H3K4me3 genomic locations was highly conserved in response to MR (Fig. 1b, c) while an overall reduction in H3K4me3 in peak intensity also observed (Supplementary Fig. 1b, c). Genes with gained or lost H3K4me3 peaks in response to MR tended to be observed in smaller peaks (Wilcoxon rank-sum P-value < 1e-4 for five out of the six comparisons, Supplementary Fig. 1d, e) and were not significantly enriched with specific biological functions compared to randomly chosen gene sets (median(−log$_{10}$(enrichment Q-value)) = 4.25 for gained peaks compared to 7.76, 8.05, and 8.98 for three random gene sets of identical size, Supplementary Fig. 1f, and median(−log$_{10}$(enrichment Q-value)) = 3.00 for lost peaks compared to 7.12, 6.05, and 5.08 for three random gene sets with identical size, Supplementary Fig. 1g) and thus likely attributable to technical noise. We also analyzed the composition of genomic elements covered by the peaks and found, as has been reported with H3K4me3[38], that most peaks (11,210 peaks, 81%) covered promoter regions with a smaller subset of peaks (2599 peaks, 19%) that appeared on non-promoter regions such as intergenic regions and introns (Supplementary Fig. 1h). To further investigate these changes, we next developed several quantitative descriptors for individual peaks (Fig. 1d). We used height, area, and width to evaluate the size of the peaks and compared these metrics in high and low methionine conditions. Genomic regions in the high and low methionine conditions were merged (Fig. 1a) to include less abundant peaks called only in one condition in the analysis. All three peak size descriptors showed high correlations (Spearman's rank correlation coefficient >0.95 and random permutation test P-value < 1e-323) between high and low methionine conditions (Fig. 1e), implying that the overall H3K4me3 landscape (i.e., relative size of each individual peak) is robust to methionine availability (linear regression coefficient = 0.87 for width, 0.85 for height, and 0.70 for area).

To determine whether this conservation in H3K4me3 dynamics under MR extends to normal physiology and under a longer-term alteration in nutrient availability that has beneficial health effects, we investigated the effects of MR on H3K4me3 in vivo. Under these conditions, MR has been shown to alter methionine metabolism, improve metabolic physiology in liver, extend life-span, and reduce global levels of H3K4me3[22,39–41]. Thus, this mouse model of liver physiology in combination with the human cancer cell model provides a breadth of model systems

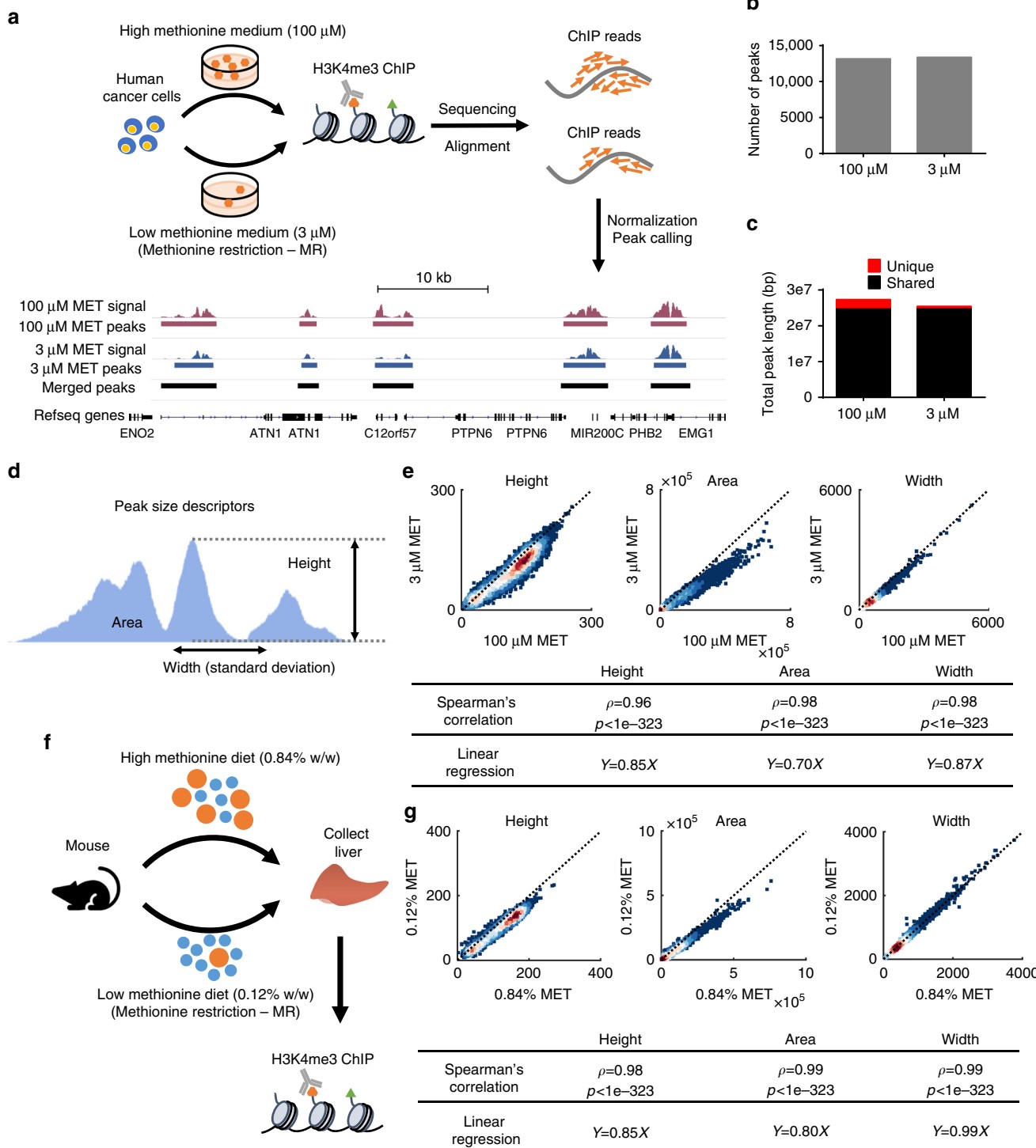

**Fig. 1** MR reduces H3K4me3 but maintains its genomic distribution. **a** MR experiment and ChIP-seq data analysis pipeline in human cancer cells HCT116. **b** Number of H3K4me3 peaks called under high (100 μM) and low (3 μM) methionine conditions in human cancer cells. **c** Total length of shared and unique peak regions for high and low methionine conditions in human cancer cells. **d** Definition of peak height, area, and width. **e** Density scatter plots comparing H3K4me3 peak heights, areas, and widths between high and low methionine conditions in human cancer cells. Colors represent for dot density. Dashed lines show identical x and y coordinates. Spearman's rank correlation coefficients between the high and low methionine conditions and linear regression coefficients are shown in the table at bottom. **f** MR experiments scheme in vivo. Mouse image used with permission from Microsoft. **g** Density scatter plots comparing H3K4me3 peak heights, areas, and widths in mouse liver between high (0.84% w/w) and low (0.12% w/w) methionine diets

covering both short-term and long-term nutrient alterations, both in vitro and in vivo systems, and both pathological and healthy contexts. We focused on liver in profiling the epigenomics and transcriptomics because it is the metabolic organ which is most responsive to metabolic reprogramming and there are liver-related phenotypes associated with MR related to metabolic health[40]. Livers were obtained from C57BL6 adult mice on a diet with either high methionine (0.84% w/w) or low, MR methionine

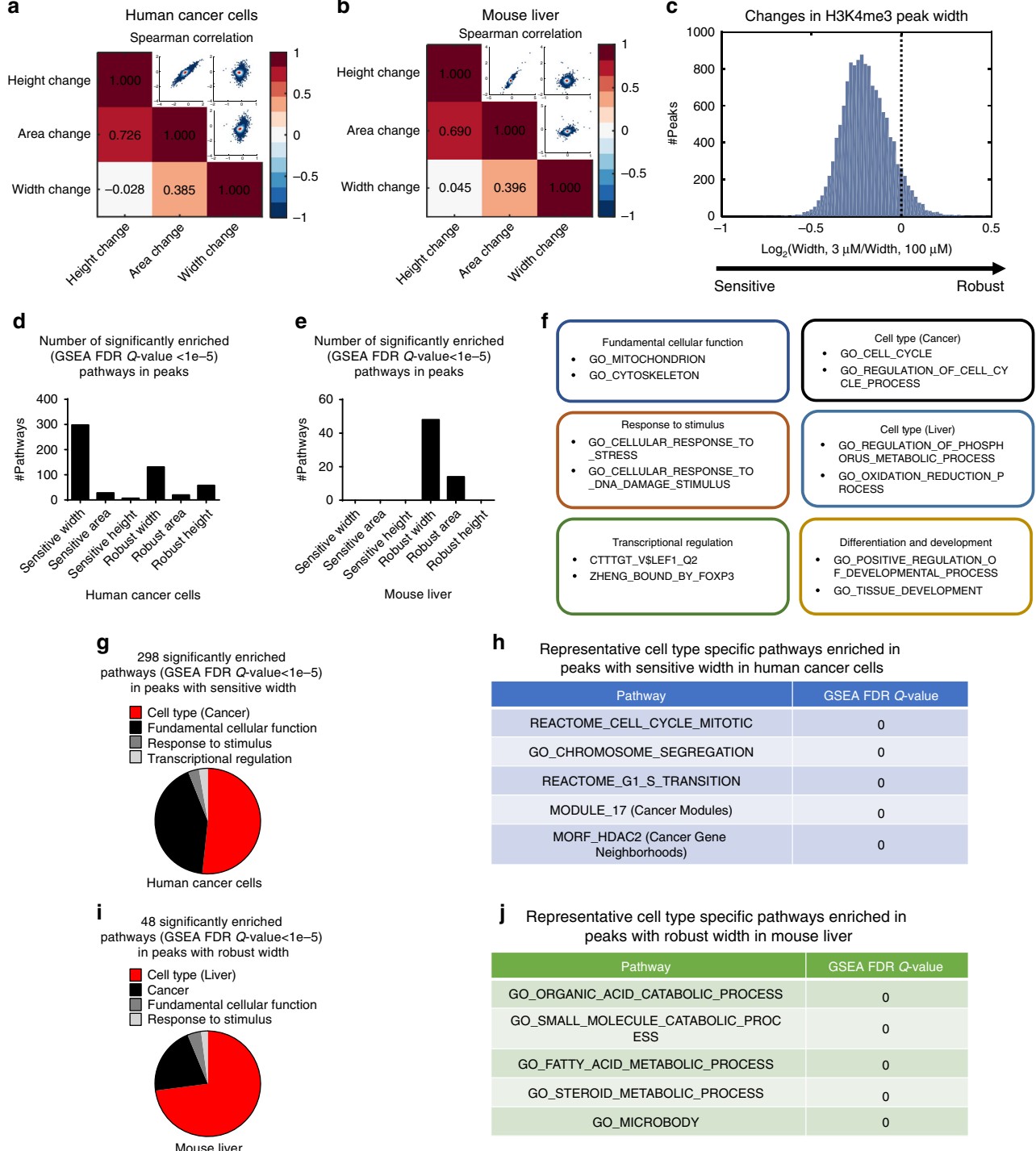

**Fig. 2** H3K4me3 width dynamics encode biological information. **a** Heat map showing Spearman's rank correlation coefficients among fold changes of H3K4me3 peak heights, areas, and widths under MR in human cancer cells. The upper right part shows density scatter plots comparing the corresponding log2(fold change) values. Colors of dots in the scatter plots indicate density of dots. **b** Same as in **a** but for mouse liver. **c** Distribution of fold changes (defined by the values in low methionine condition divided by the corresponding values in high methionine condition) in peak width. The arrow at the bottom denotes the change from sensitive (more reduction in peak width) to robust (less reduction in peak width). **d** Number of pathways significantly enriched (GSEA FDR Q-value < 1e-5) in peaks with different dynamics under MR in human cancer cells. **e** Same as in **d** but for mouse liver. **f** Examples of annotated pathways in each category. **g** Annotation of 298 MSigDB pathways enriched in H3K4me3 peaks with sensitive width in human cancer cells. **h** Representative cancer-related pathways enriched in H3K4me3 peaks with sensitive width in human cancer cells. **i** Annotation of 48 MSigDB pathways enriched in H3K4me3 peaks with robust width in mouse liver. **j** Representative liver-specific pathways enriched in H3K4me3 peaks with robust width in mouse liver

(0.12% w/w) for 12 weeks, and ChIP-seq of H3K4me3 was conducted (Fig. 1f). Quantitation of bulk levels of H3K4me3 under these two conditions confirmed as was previously published that this level of dietary MR was sufficient to reduce the levels of H3K4me3[22]. This physiological MR also revealed that the overall distribution of H3K4me3 was conserved between the different nutrient conditions as corroborated by conserved peak numbers (CV = 0.06, Supplementary Fig. 2a), location (Jaccard index = 0.82, Supplementary Fig. 2b), and marked genes (Jaccard index = 0.90, Supplementary Fig. 2c). High correlations in the values of peak height, area, and width between high methionine and low methionine conditions were also observed with peak width as the most conserved (Fig. 1g, linear regression coefficient = 0.99 for width compared to 0.85 for height and 0.80 for area). Consistent with what was found in human cancer cells, H3K4me3 peaks gained or lost in response to MR in mouse liver also tended to be significantly smaller peaks than those conserved in both high and low methionine conditions (Wilcoxon rank-sum P-value < 1e-3 for five out of the six comparisons, Supplementary Fig. 2d, e) and were not enriched with specific biological function compared to randomly chosen genes containing H3K4me3 (median($-\log_{10}$(enrichment Q-value)) = 3.09 for gained peaks compared to 4.03, 4.76, and 3.39 for three random gene sets with identical size, Supplementary Fig. 2f, and median($-\log_{10}$(enrichment Q-value)) = 14.62 for lost peaks compared to 16.58, 15.75, and 13.27 for three random gene sets with identical size, Supplementary Fig. 2g). Reduction in the overall H3K4me3 signal was also observed (Supplementary Fig. 2h, i) as was a conserved distribution of genomic elements (84% promoter peaks and 16% non-promoter peaks, Supplementary Fig. 2j).

To assess the robustness of the quantitative peak descriptors to variation in ChIP-seq data analysis methodologies, we repeated the calculations using several different peak-calling algorithms. Although the total number of peaks (CV = 0.12 for human cancer cells, Supplementary Fig. 3a, and CV = 0.24 for mouse liver, Supplementary Fig. 3b) and total length of genomic regions covered by peaks (Jaccard index = 0.60 for human cancer cells, Supplementary Fig. 3c, and Jaccard index = 0.43 for mouse liver, Supplementary Fig. 3d) exhibited some variation among methods, reads mapped to peaks (CV = 0.0073 for human cancer cells, Supplementary Fig. 3e, and CV = 0.0078 for mouse liver, Supplementary Fig. 3f), genes associated with peaks (Jaccard index = 0.78 for human cancer cells, Supplementary Fig. 3g, and Jaccard index = 0.66 for mouse liver, Supplementary Fig. 3h), and dynamics of peak height, area, and width (average Spearman's rank correlation coefficient = 0.92 for human cancer cells and 0.86 for mouse liver, random permutation test P-values < 1e-323 for all comparisons, Supplementary Fig. 4) were concordant, implying that quantitation of peak dynamics was robust to the ChIP-seq data analysis method used. Taken together, these results indicate that changes in metabolism mediated by methionine availability, despite altering global levels of histone methylation, do not induce a complete redistribution of H3K4me3 marks on the genome.

**H3K4me3 width dynamics encode biological information**. Although the genomic positioning of H3K4me3 was conserved in response to changes in methionine availability, we observed that peak height, area, and width were affected to different extents. To further investigate these differences, we computed the Spearman's rank correlation coefficients between fold changes in peak height, area, and width for both HCT116 cells and mouse liver and found that fold changes in the width and in the other two parameters were less correlated (Spearman's rank correlation coefficient < 0.4) in both models (Fig. 2a, b) relative to other comparisons. The

lower correlation between changes in peak width and changes in the other two peak features suggested that a change in peak width might encode a different dimension of information. To test this hypothesis, we conducted pathway enrichment analysis with the Gene Set Enrichment Algorithm (GSEA)[42] on the peaks with sensitive (larger reduction under MR) and robust (smaller reduction under MR) height, area, and width (Fig. 2c) using the Molecular Signatures Database (MSigDB)[43]. For each category of peak dynamics, the number of significantly enriched (GSEA FDR Q-value < 1e-5) pathways was used to quantify association of this category with biological functions. In cultured cancer cells, the change in peak width exhibited the strongest signal for enrichment of specific biological processes as measured by the number of significantly enriched pathways (298 pathways significantly enriched in sensitive width compared to 131 in robust width and less than 60 in all other categories, Fig. 2d). In mouse liver, the consistency of the peak width exhibited the strongest signal (48 pathways significantly enriched in robust width compared to 14 in robust area and 0 in all other categories, Fig. 2e). This surprising finding appears to indicate that biological information that occurs in response to changes in methionine availability is encoded in H3K4me3 peak width.

To further quantify the information contained in peak width, we annotated the significantly enriched pathways in each case by classifying them into categorical subsets involving fundamental cellular function, response to stimulus, cell type (cancer, proliferation-related pathways for human cancer cells and metabolism, multicellular organ-related pathways for mouse liver), transcriptional regulation, differentiation and development, and cancer (Fig. 2f). We next computed the fraction of each category in the pathways enriched in sensitive width in cancer cells and those enriched in robust width in liver. Unexpectedly, these two sets were enriched with cell type-specific biological functions. Proliferation and cancer-related pathways were enriched in peaks with sensitive width in cancer cells (154 out of the 298 enriched pathways, Fig. 2g, h), while metabolism and multicellular organ-related pathways were enriched in peaks with robust width in mouse liver (35 out of the 48 enriched pathways, Fig. 2i, j). Together these findings indicate that the peak width dynamics in response to MR encodes information about biological function.

**H3K4me3 width dynamics encode cell type-specific TF binding**. To further explore this relationship, we probed additional aspects of genomic architecture. Using a motif analysis algorithm, HOMER[44], we searched for transcription factor (TF) binding motifs enriched in the subsets of the different geometrical features of H3K4me3 peaks that change or remain consistent during MR. Peak subsets with the 500 most sensitive peaks or 500 most robust peaks were considered in this analysis (Fig. 3a). The complete set of H3K4me3 peaks was used as the background model (Fig. 3b). In cultured cancer cells, TF binding motifs were only enriched (one-sided Fisher's exact test Q-value < 0.05) in peaks with sensitive width (78 motifs enriched in sensitive width compared to 1 for robust area, 2 for robust width, and 0 for all other peak sets, Fig. 3c, Supplementary Fig. 5a), while in mouse liver TF binding motifs were only enriched in peaks with robust width (249 motifs enriched in robust width compared to 0 for all other peak sets, Fig. 3d, Supplementary Fig. 5b). These findings were also found to be robust to the size of peak subsets chosen (Supplementary Fig. 5c, d). These two sets of TF binding motifs have little overlap (Jaccard index = 0.05 between the two sets of 10 motifs with smallest enrichment Q-values), suggesting that the role of H3K4me3 dynamics in relation to TF binding has a tissue-specific function. The top scoring TF motifs in cancer cells that associated with sensitive H3K4me3 peak width dynamics tended to involve cancer-

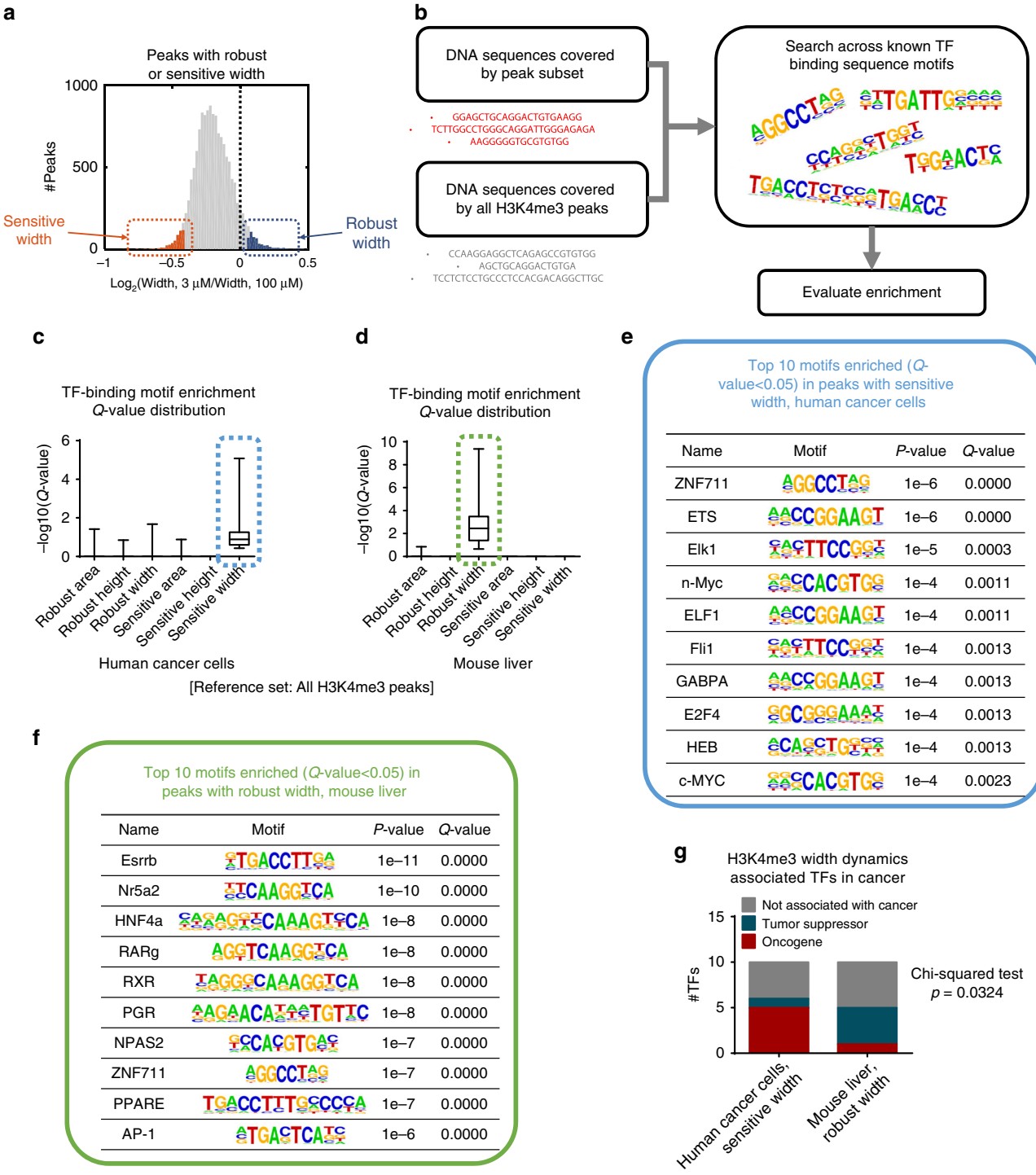

**Fig. 3** H3K4me3 width dynamics encode cell type-specific TF binding. **a** Distribution of fold changes (defined by the values in low methionine condition divided by the corresponding values in high methionine condition) in peak width in all peaks (gray), top 500 peaks with sensitive width (orange) and top 500 peaks with robust width (blue) in human cancer cells. **b** Framework for the TF binding motif enrichment analysis. **c** Distributions of TF binding motif enrichment $Q$-values in H3K4me3 peak sets with different dynamics under MR in human cancer cells. Box limits are the 25th and 75th percentiles, center lines are medians, and the whiskers are the minimal and maximal values. **d** Same as in **c** but for mouse liver. **e** Top 10 TF binding motifs enriched in 500 H3K4me3 peaks with sensitive width in human cancer cells. **f** Same as in **e** but for robust width in mouse liver. **g** Number of oncogenes and tumor suppressors in top 10 TFs enriched in H3K4me3 peaks with sensitive width in human cancer cells and those with robust width in mouse liver

associated TFs such as c-MYC and ETS in cancer cells (Fig. 3e) and liver-specific TFs such as RXR and ESRRB in liver (Fig. 3f).

To investigate the binding of these TFs, we obtained ChIP-seq datasets for two TFs putatively associated with H3K4me3 width in each system (Supplementary Fig. 6a) and computed the overlap between the TF binding sites and H3K4me3 peak subsets. We also chose one TF without binding motifs associated with peak width for each system as a negative control (Supplementary Fig. 6b). We found that the cancer-related TFs, MYC (Supplementary Fig. 6c), and ELF1 (Supplementary Fig. 6d) indeed had

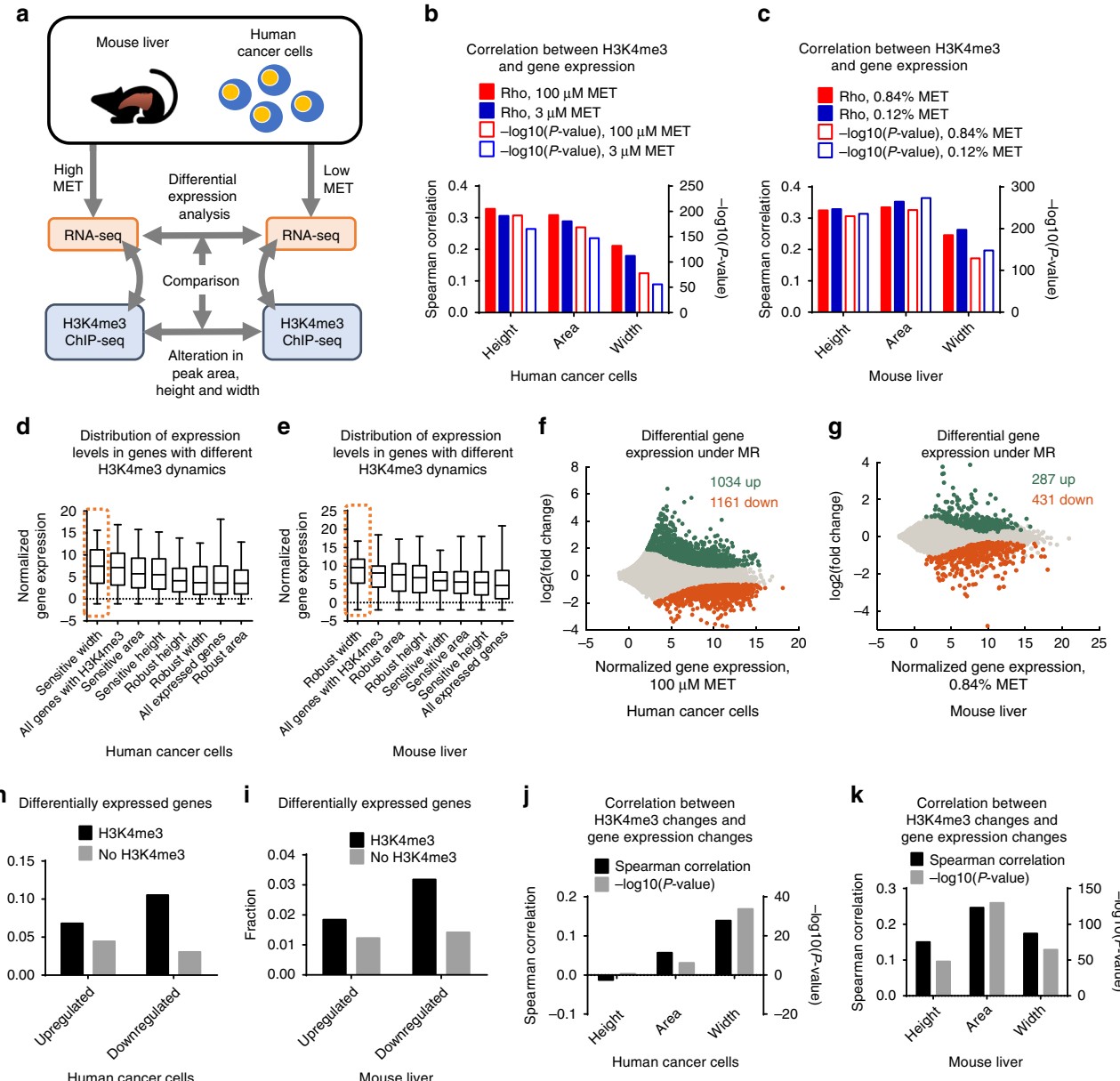

**Fig. 4** H3K4me3 width dynamics predict differential gene expression. **a** Framework of RNA-seq data analysis. Mouse image used with permission from Microsoft. **b** Spearman correlation between peak size descriptors and gene expression levels in human cancer cells. **c** Same as in **b** but for mouse liver. **d** Expression levels of genes associated with different H3K4me3 dynamics under high methionine conditions in human cancer cells. The gene set with sensitive width which has been demonstrated to associate with cell type-specific biological functions and TF binding is highlighted. Box limits are the 25th and 75th percentiles, center lines are medians, and the whiskers are the minimal and maximal values. **e** Same as in **d** but for mouse liver. **f** Differential gene expression in human cancer cells under MR. **g** Same as in **f** but for mouse liver. **h** Fraction of differentially expressed genes in genes with or without H3K4me3 in human cancer cells. **i** Same as in **h** but for mouse liver. **j** Spearman's rank correlation coefficients between H3K4me3 changes and gene expression changes under MR in human cancer cells. **k** Same as in **j** but for mouse liver

binding sites enriched (one-sided Fisher's exact Q-value < 1e-10) in peaks with sensitive width in human cancer cells, while the liver-specific TFs, HNF4a (Supplementary Fig. 6e), and RXRa (Supplementary Fig. 6f–h) were also enriched (one-sided Fisher's exact Q-value < 1e-10) in peaks with robust width in mouse liver. Such patterns were not observed in the TFs used as negative controls (one-sided Fisher's exact Q-value = 1, Supplementary Fig. 6i, j). We also observed the strongest correlation between width dynamics and the number cell type-specific TFs bound to the genomic region containing the H3K4me3 peak (one-way ANOVA P-value = 3.53e-319 for width changes in human cancer cells and 2.98e-189 for width changes in mouse liver,

Supplementary Fig. 7). This concordance suggests that cell identity is encoded in the H3K4me3 width dynamics but not in any other structural or locational aspect of H3K4me3. A previous study has found that broad H3K4me3 peaks are associated with tumor suppressor gene function in normal cells[34]. We also found that TFs with binding motifs enriched in preserved H3K4me3 peak width in liver tended to be tumor suppressors, while those enriched in sensitive H3K4me3 peak width in cancer cells tended to be oncogenes (chi-squared test P-value = 0.0324, Fig. 3g), suggesting that H3K4me3 width dynamics under MR may also encode information about tumor suppression or progression, depending on the cell or tissue type. Altogether, these analyses

highlight H3K4me3 peak width dynamics as the information carrier under MR in both human cancer cells and mouse liver.

**H3K4me3 width dynamics predict differential gene expression.**
Despite a well-known association of the H3K4me3 mark with active promoters, the exact role of H3K4me3 in mediating gene expression is a matter of current debate[45,46]. We therefore asked whether any aspects of genomic alterations in H3K4me3 track with changes in gene expression under MR. We first quantified gene expression levels using RNA sequencing (RNA-seq) under high and low methionine conditions and correlated the gene expression levels with H3K4me3 peak features again in both cancer cells and mouse liver (Fig. 4a). In cancer cells, we observed 7709 expressed genes and 5118 non-expressed genes marked by H3K4me3 (Supplementary Fig. 8a). There were also 11,412 expressed genes without H3K4me3 marks, indicating in our models that the presence of the H3K4me3 mark is neither necessary nor sufficient for gene expression. Nevertheless, expressed genes with no H3K4me3 had significantly lower expression levels (Wilcoxon rank-sum $P$-value < 1e-323, Supplementary Fig. 8b), supporting the well-characterized association between H3K4me3 and active gene expression. This observation was further corroborated by significantly smaller H3K4me3 peak sizes in non-expressed genes (Wilcoxon rank-sum $P$-values < 1e-26 for height, area, and width, Supplementary Fig. 8c–e). An evaluation of the correlation coefficients between H3K4me3 peak height, area, width, and gene expression levels revealed significant positive correlations (Spearman's rank correlation coefficient > 0.2, random permutation test $P$-value < 1e-50) between all three peak size descriptors and gene expression levels in both high and low methionine conditions (Fig. 4b, Supplementary Fig. 8f). Consistently, in mouse liver, we identified 9417 expressed and 2449 non-expressed genes with H3K4me3, as well as 9282 expressed genes without H3K4me3 (Supplementary Fig. 9a). In accordance with our findings in cultured human cells, strong correlations between the presence of H3K4me3 and gene expression levels were also observed in liver ($P$-value < 1e-20 for all Wilcoxon rank-sum and random permutation tests, Fig. 4c, Supplementary Fig. 9b–f). Thus, in our models, the presence of an H3K4me3 peak, while not a requirement for gene expression, predicts overall whether a gene is expressed, and the magnitude of the peak does appear to contain information about overall gene expression level. This finding is consistent with the well-known association between H3K4me3 around transcription start sites (TSS) and active transcription[47].

Having established a baseline for the relationship between H3K4me3 and gene expression, we next sought to study whether changes in the geometrical features of H3K4me3 are connected to changes in gene expression. We compared expression levels of genes associated with different H3K4me3 dynamics in cancer cells (Fig. 4d) and liver (Fig. 4e) under high methionine conditions. In addition to our previous findings that sensitive H3K4me3 peak width in cancer cells and robust H3K4me3 peak width in mouse liver are indicative of biological function, we found that genes associated with these peaks also exhibit significantly higher expression levels (Wilcoxon rank-sum and Kolmogorov–Smirnov $P$-values < 0.05, Fig. 4d, e, Supplementary Fig. 10). We then conducted differential expression analysis to identify the genes with altered expression in response to MR. In human cancer cells, we found 1034 genes upregulated (Wald test $Q$-value < 0.05, $\log_2$(fold change) > 0) and 1161 genes downregulated (Wald test $Q$-value < 0.05, $\log_2$(fold change) < 0) under MR (Fig. 4f) and 287 upregulated (Wald test $Q$-value < 0.05, $\log_2$(fold change) > 0) and 431 downregulated (Wald test $Q$-value < 0.05, $\log_2$(fold change) < 0) genes in mouse liver (Wald test $Q$-

value < 0.05) (Fig. 4g) in diverse classes of genes. It is noteworthy that differential gene expression in vivo is confounded by factors including composition of liver by different cell types and larger variation between individual mice. Thus, as expected fewer genes were found to be differentially expressed in mouse liver.

We next asked if changes to or consistencies in the geometrical features of H3K4me3, especially in peak width, can predict changes in gene expression. We first compared the fraction of differentially expressed genes that contain or are absent of H3K4me3. We found that in both cancer cells and liver, H3K4me3-marked genes were enriched in both upregulated and downregulated genes (one-sided Fisher's exact $P$-value = 1.82e-12 for upregulated genes and 1.89e-98 for downregulated genes in human cancer cells, Fig. 4h, and 4.22e-4 for upregulated genes and 4.43e-16 for downregulated genes in mouse liver, Fig. 4i), suggesting that the presence of H3K4me3 notes a tendency of differential expression during MR. Next, we correlated fold changes in H3K4me3 peak height, area, width with changes in gene expression levels and found that H3K4me3 peak width dynamics significantly correlated (random permutation test $P$-value < 0.05, Spearman's rank correlation coefficient > 0.1) with alterations in gene expression levels, and this correlation was consistent in both models (Spearman's rank correlation coefficient = 0.14 for human cancer cells and 0.17 for mouse liver, Fig. 4j, k). On the other hand, although H3K4me3 peak height and area dynamics were also found to correlate with differential gene expression in mouse liver (random permutation test $P$-value < 0.05, Spearman's rank correlation coefficient > 0.15, Fig. 4k), the strength of these correlations was smaller in human cancer cells (Spearman's rank correlation coefficient < 0.06, Fig. 4j). Restricting the analysis to the subset of peaks located at promoters (Supplementary Fig. 11a) or altering the data analysis pipeline (Supplementary Fig. 11b) had minimal effects on the resulting correlation coefficients (absolute differences in Spearman's rank correlation coefficients < 0.04 for data in Fig. 4j, Supplementary Fig. 11a, b). To further assess the predictability of changes in gene expression from changes in peak height, area, and width, we performed multiple linear regressions with changes in peak height, area, and width as the independent variables and changes in gene expression as the dependent variable. Change in peak width is the only variable with significant non-zero linear coefficients in predicting changes in gene expression in both human cancer cells and mouse liver ($P$-value = 3.75e-15 in human cancer cells and 8.39e-7 in mouse liver, Supplementary Fig. 11c, d). Interestingly, we also observed a stronger correlation between H3K4me3 width changes and gene expression changes in genes with H3K4me3 peaks bound by more TFs putatively associated with H3K4me3 width in both human cancer cells (Spearman's rank correlation coefficient between width changes and gene expression changes = 0.19 in peaks bound by 2 TFs compared to 0.10 for 0 TF and 0.12 for 1 TF, Supplementary Fig. 11e) and mouse liver (Spearman's rank correlation coefficient between width changes and gene expression changes = 0.22 in peaks bound by 2 TFs compared to 0.18 for 0 TF and 0.19 for 1 TF, Supplementary Fig. 11f), suggesting that TFs associated with H3K4me3 width dynamics regulate expression of their target genes thus mediating the connection between H3K4me3 dynamics and gene expression dynamics. Moreover, peaks with sensitive width in human cancer cells tended to be associated with downregulated genes (one-sided Fisher's exact test $P$-value = 0.02, Supplementary Fig. 11g), which were also enriched with more cancer-related pathways (305 pathways among which 188 were cancer-related enriched in downregulated genes compared to 2 pathways enriched in upregulated genes, Supplementary Fig. 11h, i), suggesting that the correlation between H3K4me3 width dynamics and differential gene expression is linked to

biological outcomes under MR. Similar pattern was also observed in mouse liver, in which upregulated genes were associated with peaks with robust width and enriched with liver-specific functions (Supplementary Fig. 11j–l). Taken together, H3K4me3 width dynamics, but not area or height, is the predictor of alterations in gene expression in both human cancer cells and mouse liver. Moreover, peak width also encodes information of gene expression levels under normal methionine conditions in addition to cell type-specific biological functions and TF binding preferences.

## Discussion

Numerous studies have shown that global levels of histone and DNA modifications are influenced by metabolism and changes to nutrient availability[1–4]. Biological outcomes, however, result from reprogramming of chromatin state which influences regulation of gene expression and these mechanisms are still poorly understood. This study possibly identifies general principles about how specific aspects of the genomic architecture of a histone mark are affected by nutrient availability.

We found that H3K4me3, a chromatin mark known to associate with active transcription, responds to MR with a global compression of peak area and height across most modified sites, which is consistent with the substantial reduction of bulk levels observed previously[22,23]. H3K4me3 peak width, despite being the most conserved feature under MR, uniquely encoded, in its dynamics, information about cell identity, TF binding preferences, tumor suppression, and gene expression that are not reflected in changes in other aspects of H3K4me3. Peak width dynamics was also the only predictor of gene expression alterations in both human cancer cells and mouse liver. In addition to peak width encoding information about cell identity[33,34], we showed that its dynamics under alterations in methionine metabolism is also a link between H3K4me3 and changes to gene expression. This finding extends our understanding of how metabolism influences chromatin biology since it identifies aspects of biological specificity in H3K4me3 dynamics across the genome.

Finally, we observed opposing effects in cultured human cancer cells and in mouse liver. H3K4me3 width changes in cancer cells and width conservation in liver marked biological functions and gene expression changes. In both cases, the biological associations with these dynamics may be attributed to the function of the tissue and the physiological or pathophysiological status of the model in relation to the ongoing gene expression programs. The discrepancy between cancer cells and healthy tissue may also reflect differences between cancerous and normal cell types in responding to alterations in environmental factors. We speculate that, at least in the context of MR, cell type-specific functions in normal tissues are more robust under alteration in environmental variables to ensure normal function, while cancer cells have more flexibility which potentially maximizes fitness in a varying environment. Notably, we found that genes related to proliferation and survival of HCT116 cancer cells identified in a CRISPR screen[48] exhibited significantly elevated sensitivity in both H3K4me3 width (Wilcoxon rank-sum $P$-value = 1.47e-136) and gene expression (Wilcoxon rank-sum $P$-value = 8.15e-116) in response to MR (Supplementary Fig. 12), suggesting that the influences of MR on H3K4me3 peak width are indeed linked to functional outcome. Although further investigation is needed to unravel the mechanism conferring this discrepancy between pathological and healthy models, the general principle that the peak width is the most informative parameter in H3K4me3 dynamics upon changes in nutrient availability is conserved in both models.

Although long appreciated to be a signature of active gene expression[47], the role of H3K4me3 in regulation of gene expression remains controversial[46]. There is evidence that H3K4me3 interacts with transcriptional and splicing machineries to regulate gene expression[49,50], while other studies have concluded that the timing of H3K4me3 changes across biological conditions precludes it having an active role in gene expression[51,52]. Other studies have concluded that the mark may affect the robustness of transcriptional programs[33]. Recent studies of H3K4me3 during early embryo development have further suggested that the function of H3K4me3 in this process may be to counteract DNA methylation at specific genomic regions[35,37], supporting a function independent of facilitating transcription. In this study, we showed that reprogramming of the H3K4me3 landscape under an alteration in nutrient availability was indeed related to alterations in gene expression but through a specific mechanism involving the dynamics of H3K4me3 peak width. Although more work is needed to establish definitive causality between H3K4me3 and gene expression dynamics, our studies support a model that a subset of overall gene expression appears to be responsive to changes in H3K4me3 and that these programs are encoded in changes in peak width.

There remain numerous unanswered questions on the interaction between metabolism and chromatin biology. Chromatin status is a manifestation of more than 100 covalent modifications and multiple assembly factors[53,54] and numerous metabolic pathways beyond one-carbon metabolism directly interact with chromatin. It is to be determined how general are the principles we found regarding H3K4me3 peak width and whether they extend to other marks or to other metabolic changes that alter methylation such as those regulated by mitochondrial metabolism and alpha-ketoglutarate. For example, H3K4me3 levels are also reduced by knockdown of the histone methyltransferase MLL1 in HCT116 cells[55], but the conserved and unique features in this process compared to MR is not clear, although there is existing literature that has defined some aspects of the specificity of the requirements of methyltransferases[56–58]. In conclusion, this study may define principles of how metabolism influences chromatin biology, that is, almost all aspects of H3K4me3 biology, including cell identity, tumor suppression and progression, and gene expression are encoded in H3K4me3 width dynamics.

## Methods

**Methionine restriction in human cancer cells and mouse.** HCT116 cells were obtained from ATCC and the stock used for this study was recently validated as bona fide HCT116 cells via the Duke University DNA Analysis Facility Human cell line authentication service and validated to be mycoplasma free. Cells were cultured in RPMI with 10% FBS (containing 30 micromolar methionine). Plated cells were first cultured in 100 micromolar methionine and switched to 3 micromolar for 24 h upon harvesting lysates. Seven-week-old male C57BL/6J mice were randomized and fed with either 0.84% (w/w) methionine control diet or 0.12% (w/w) methionine MR diet for 12 weeks and fasted before being sacrificed. Mice were randomized to minimize difference in body weight between the control and MR groups. No blinding was used. All animal procedures were approved by the Institutional Animal Care and Use Committee of the Orentreich Foundation for the Advancement of Science (Permit Number 0511MB). Input chromatin was pooled from all samples in each replicate.

**H3K4me3 ChIP-seq data analysis.** ChIP was performed using $1.5 \times 10^7$ cells and the Millipore ChIPAb + H3K4me3 antibody (cat #17G614, lot #2196044, dilution 3:500) with Protein A Agarose beads (Millipore, cat #16G125, lot #2444123). In addition, we applied a spike-in normalization strategy for generating quantitative ChIP-seq data[59], in which spike-in Drosophila chromatin and spike-in antibody for Drosophila H2Av (Active Motif cat #61686, dilution 1:250, 2 μg total) was mixed with chromatin from the HCT116 cells before the chromatin IP step with a fixed ratio (2:1, Drosophila:HCT116 chromatin) as a reference. Libraries were prepared according to Illumina instructions and sequenced on the Illumina HiSEQ 2500 sequencer in Rapid Run Mode at the Duke GCB Sequencing Shared Resource. Reads were aligned to a combinational genome consisting of the human reference genome hg19 and the Drosophila reference genome dm6 using Bowtie2[60].

Alignment files were then filtered according to their alignment scores and down-sampled to ensure each file contains the same amount of unique Drosophila reads. Finally, reads mapped to hg19 in the normalized alignment files were kept for peak-calling and other following analyses. Replicate 1 for high methionine condition was not used due to abnormally high fraction of Drosophila-originated reads in this sample. Processing of alignment files was done using SAMtools[61]. Peaks were called using the --broad mode of MACS2[62] and filtered with the criterion that enrichment Q-value is smaller than $10^{-5}$. For mouse liver, ChIP-seq reads were aligned to the mouse reference genome mm8 and normalized by sequencing depth.

**Peaks annotation and size descriptors computation**. H3K4me3 peak regions called for different methionine conditions and replicates were merged using the bedtools merge command in BEDTools package[63] to generate a combinational peak set for following annotation and computation of peak descriptors. Peaks were assigned to genes with TSS closest to center of the peak region using Homer[44]. Peak height, area, and width were computed on this combinational peak set using the extended reads coverage files (that is, number of fragments extended from ChIP-seq reads mapped to each base pair on the genome) generated by MACS2. Height was computed by searching for position with highest read coverage. Area was computed by integrating the reads coverage file over the peak region. Width was quantified by first normalizing reads coverage profile of each peak to a probability distribution function (i.e., area under the coverage curve = 1) and then calculating standard deviation accordingly. Percentage peak areas on each type of genomic elements were computed using C++ codes with genomic element annotation files in the HOMER package and reads coverage files generated by MACS2 as inputs. Replicates were merged by computing average between them.

**Comparison between peak calling methods**. H3K4me3 peaks were called using Bayesian Change Point[64], MUltiScale enrIchment Calling for ChIP-seq[65], and MACS2 either with (MACS2.broad) or without (iMACS2.narrow) the -broad option. Default parameters for each of these methods were used. Generation of unions and intersections of the peak sets and quantification of reads mapped to peak regions was conducted with BEDTools using the commands bedtools merge, bedtools intersect, and bedtools coverage. Extended read coverage profiles were generated using MACS2 as described previously. Raw read coverage profiles were generated from the normalized alignment files using the command bedtools gen-omecov in BEDTools without applying the model in MACS2 for extension of reads to whole fragments. For the comparison of H3K4me3 changes between peak-calling algorithms, peaks called by the two algorithms assigned to the same gene were compared with each other.

**Pathway and TF binding motif enrichment analysis**. Pathway enrichment analysis with GSEA was done using the tool "Run GSEAPreranked" in the javaGSEA Desktop Application (http://software.broadinstitute.org/gsea/downloads.jsp). Gene sets H (hallmark gene sets), C1 (positional gene sets), CP (canonical pathways), CP:BIOCARTA (BioCarta gene sets), CP:KEGG (KEGG gene sets), CP:REACTOME (Reactome gene sets), C3 (motif gene sets), C4 (computational gene sets), and C5 (GO gene sets) in the MSigDB 6.0 (http://software.broadinstitute.org/gsea/msigdb/index.jsp) were included in the analysis. Genes were ranked in ascending order according to changes in height, area, or width of H3K4me3 peaks associated with them (for pathway enrichment analysis in genes with different H3K4me3 dynamics) or according to the Wald statistic (for pathway enrichment in differentially expressed genes). The ranked lists of genes were used as input to the GSEA algorithm with the parameter "enrichment statistic" set to "classic". For all other parameters, the default values were used. TF binding motif enrichment analysis was done by the module findMotifsGenome.pl in HOMER. Lists of tumor suppressors and oncogenes were obtained from the databases TSGene 2.0[66] and ONGene[67].

**TF ChIP-seq data analysis**. Raw read files in SRA or FASTQ format for the TF ChIP-seq experiments were downloaded using the accession numbers in Supplementary Fig. 6A and B, aligned to the reference genome hg19 for HCT116 cells and mm8 for mouse liver using Bowtie2, and filtered according to alignment scores using SAMtools. SRA files were converted to FASTQ format using the command fastq-dump in the SRA Toolkit (https://github.com/ncbi/sra-tools) before the alignment. Peaks were called using MACS2 without the --broad option and filtered with the criterion that the enrichment Q-value is smaller than $10^{-5}$. H3K4me3 peaks bound by a TF were defined as those H3K4me3 peaks overlapping with peaks called from the corresponding TF ChIP-seq data. Fold enrichment of TF binding in a H3K4me3 peak subset was defined as the ratio of fraction of peaks bound by the TF in this H3K4me3 peak subset relative to the fraction of peaks bound by the TF in the complete set of H3K4me3 peaks, that is:

$$\text{Fold enrichment}$$
$$= \frac{\#(\text{H3K4me3 peaks bound by TF in subset})/\#(\text{H3K4me3 peaks in subset})}{\#(\text{H3K4me3 peaks bound by TF})/\#(\text{H3K4me3 peaks})}.$$

Enrichment Q-values were computed by correcting P-values from one-sided Fisher's exact test using Benjamini–Hochberg procedure.

**RNA-seq**. Total RNA from HCT116 cells and mouse liver under high and low methionine conditions was extracted using the PARIS kit (Life Technologies, cat #AM1921), polyA selected and then sent to the Weill-Cornell Epigenomics core (HCT116) and the Duke GCB Sequencing Shared Resource (mouse liver) for library preparation and sequencing. Libraries were sequenced either on the Illumina HiSEQ 2500 sequencer in Rapid Run Mode (HCT116) or on the Illumina HiSEQ 4000 sequencer (mouse liver).

**RNA-seq data analysis**. Raw reads were aligned to the human reference genome hg19 and mouse reference genome mm8, respectively, using TopHat2[68]. Number of reads mapped to each gene feature was first quantified using HTSeq[69] with the input of GTF files obtained from the UCSC Table Browser and then normalized using DESeq2[70]. Differential expression analysis was done with DESeq2. P-values were adjusted using the Benjamini–Hochberg procedure. Genes with adjusted differential expression P-values smaller than 0.05 were considered as differentially expressed.

**Fitness genes in human cancer cells**. The list of fitness genes was extracted from Table S2 of the published study on CRISPR-based screen of proliferation and survival-related genes in HCT116 cells[48].

**Statistical analysis**. The sample sizes were selected to enable evaluation of statistical significance of difference between groups. Comparison between replicates was performed before the following analysis to ensure that the variance was similar between the groups. For ChIP-seq, we performed two biological replicates in human cancer cells and two technical replicates in mouse liver. For RNA-seq, we performed two biological replicates in human cancer cells and six biological replicates in mouse liver. For multiple hypothesis testing, the P-values were adjusted using the Benjamini–Hochberg procedure. P-value < 0.05 or Q-value < 0.05 was considered statistically significant.

**Data visualization**. Heatmaps and average profiles showing the H3K4me3 ChIP-seq signal were generated with the commands plotHeatmap and plotProfile in deepTools. Other heatmaps and density scatter plots were created using MATLAB. All bar graphs, pie graphs, and box plots were created using GraphPad Prism. In all box plots, the boxes extend between the 25th and 75th percentiles, the center line shows the median, and the whiskers represent the minimal and maximal values. ChIP-seq tracks were created using Integrative Genomics Viewer (http://software.broadinstitute.org/software/igv/).

**Code availability**. Codes are available at GitHub page of the Locasale Lab (https://github.com/LocasaleLab/H3K4me3_MET_Restriction).

**Data availability**. Processed data are available at GitHub page of the Locasale Lab (https://github.com/LocasaleLab/H3K4me3_MET_Restriction). Raw data are available at the Gene Expression Omnibus database with accession number GEO: GSE103602. Peak height, area, width, and gene expression values in different conditions are available in Supplementary Data 1.

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

## Acknowledgements

Support from the American Cancer Society (RSG-16-214-01-TBE) and National Institutes of Health (R01CA193256, R00CA168997, P30CA014236) are gratefully acknowledged. Z.D. thanks Dr. Ning Yin for helpful discussions, Dr. Zhengtao Xiao for help with the GSEA analysis, and Annamarie Allen for comments on the text. We thank Dwight Mattocks (Orentreich Foundation for the Advancement of Science) for assistance with animal husbandry. Support for computational resources from the Duke Compute Cluster and Data Commons Storage is gratefully acknowledged.

## Author contributions

Z.D. and J.W.L. designed the study and wrote the manuscript with input from S.J.M.; Z. D. performed the data analysis; S.J.M. performed the experiments with help from X.G. S. N.N. performed mouse feeding and tissue harvesting.

## Additional information

**Competing interests:** The authors declare no competing interests.

