## [Peer Review File · Nature Communications]

Reviewers' comments:

Reviewer #1 (Remarks to the Author):

In this manuscript, Dai et al. studied how nutrient availability influences the genomic architecture of H3K4me3 and gene expression and function. Analysis of genome-wide distribution of H3K4me3 in human cancer cells and mouse liver in response to alternations in methionine availability revealed that H3K4me4 peak width dynamics encodes biological information including transcription factor binding, gene expression and function. This manuscript addresses a timely and interesting question and makes some interesting observations. This work will be more impactful if experimental validation of observations/predictions can be done and a link between H3K4me3 width encoded transcription factor binding and gene expression can be established.

1. Figure 1. How does methionine restriction influence cancer cell growth and survival?
2. Figure 3. Can the authors validate the binding of some putative transcription factors in human cancer cells and mouse liver?
3. Figure 4. As histone modifications need to orchestrate specific transcription factor binding to regulate transcription, can the authors perform an integrative analysis of H3K4me3 dynamics, cell-type specific transcription factor binding and gene expression?

Reviewer #2 (Remarks to the Author):

The manuscript by Dai et al is focused on the relevance of H3K4me3 peak width dynamics in regulating gene expression and, consequently, biological functions. The authors used the methionine restriction (MR) as experimental paradigm in vitro (HCT116 cells) and in vivo (liver of C57BL6 mice exposed to a control diet or to a low methionine diet). They investigated the genome-wide changes in H3K4me3, analyzing height, area and width of the ChIP-seq peaks, and concluded that peak width directly correlates with gene expression (RNA-seq) changes and stress that this parameter is linked to key biological process (acquisition of cell identity) and pathophysiological mechanisms (tumor suppression and progression).

Major comments

- The aim of the manuscript is to identify a new relevant parameter of an epigenomic analysis (the width of H3K4me3 peak) to study the regulation of gene expression and, lastly, the modulation of biological functions. To this end, the authors used an experimental paradigm of methionine restriction. As previously published by the same group (Histone Methylation Dynamics and Gene Regulation Occur through the Sensing of One-Carbon Metabolism, *Cell Metab.* 2015 Nov 3; 22(5): 861–873, Mentch et al.) the methionine restriction in HCT116 cells affect the global H3K4me3 profile. Similar to their previously published results, they showed an overall reduction in H3K4me3 in peak intensity (Figure S1E, S1F). However they did not detect any difference in the peak number between cancer cells exposed to 100uM or 3uM methionine (Figure 1B), indicating that methionine restrictions has no effect on genomic distribution of this histone modification. Thus, they decided to focus on the peak descriptors and they identify peak width as a relevant feature correlating with gene activation. Several times in the manuscript the authors reported that the peak width of ChIP seq analysis could be correlated to activation/inactivation of specific biological functions. However, based on the results they provide, it is not possible to establish a direct correlation between peak width and biological processes (except for a certain gene expression profile). To make their observation more relevant, the authors should modify the experimental paradigm. Since the authors chose a cellular model of cancer, they should demonstrate how changes in H3K4me3 peak width influences cancer cells in terms of proliferation/function, as a proof of principle that the alteration of this parameter in an epigenetic analysis really correlates with a functional output.

- To analyze the effect of long-term alterations in nutrient availability, the authors used an in vivo model, focusing on the liver. However, the rationale of the choice of this paradigm is not properly justified in the manuscript. To strengthen the take-home-message the authors should perform further experiments in an in vivo model of cancer, investigating the effect (epigenetic and functional) of methionine restriction in this model. If there are scientific reasons making this experiment not feasible, the authors should clearly justify them, and further explain why they chose to study the effect of MR in the liver.

- Results reported in figure 2 highlight that relevance of MR on peak width is different in cancer cells and in liver: the GSEA analysis showed that in HCT116 cells the change of peak width (sensitive peaks) exhibits significant enrichment of specific processes. On the other hand, in the liver the correlation is between the consistency of the peak width (robust peaks) with biological information. Which is the meaning of this result? The authors picked the liver model to investigate the long-term effect of MR and to strengthen the in vitro observations. However, the outcome is different: robust peaks in the liver are the ones correlating with enrichment of biological processes. Does this mean that under MR a set of genomic regions (that are the ones related to relevant biological processes, such as metabolism) in the liver show conserved peak width of H3K4me3? If so:

1) Is this related to the experimental paradigm used (for example exposure to the 0.84% or 0.12% methionine diet was too short)? According to this hypothesis, the number of genes modulated by MR in the liver (in terms of gene expression, figure 4G) is lower as compared to the number of differentially regulated genes in cancer cells. The use of a different in vivo model could help in resolving this discrepancy.

2) Or does it mean that differences of peak width on genes related to metabolism in liver is less dynamic compared to cancer cells and consequently more conserved in response to external stimuli, such as MR?

- This manuscript suggests a new interesting parameter that can give insights in analyzing ChIP-seq data. Do the authors think that peak width could be more relevant than the H3K4me3 enrichment itself? Do they compare the correlation between H3K4me3 and gene expression and the correlation between H3K4me3-peak width with gene expression? Do the author think this can be relevant and potentially explain why mild changes in H3K4me3 enrichment can correlate with big changes in gene expression?

Minor comments

- The authors should provide further information about the statistical analysis that has been performed. If they mention statistically significant changes they should indicate the p values in the graph and in the figure legends they should give information about the type of statistical test they used.

- I recommend a revision of the manuscript by an English native speaker.

Reviewer #3 (Remarks to the Author):

Dai et al. performed a systematic analysis of H3K4me3 peak shape and found strong associations between gene expression/pathway enrichment/TFBSs and peak width, but not peak height or peak area. In general, the observation is interesting and the conclusions are well supported by the data. There are some issues that need further clarification.

Major:

1. HCT116 cells are treated by methionine with two different concentrations. But the authors do not find much difference under the two conditions in Fig 1. How did the authors choose such two concentrations and is the 3uM sufficient to alter chromatin?

3. Sensitive width in human data is enriched in GSEA analysis, whereas Robust Width in mouse

data is enriched (Fig 2 ~ 4). The authors need to explain in detail why there is such dramatic discrepancy in two closely related species, and need to rule out the possibilities of any technical issues.

4. in Fig 3f, the association of sensitive width with TSG or robust width with OG is not convincing enough. A more rigorous statistical analysis is needed.

Minor:

1. Fig 4C is not mentioned in the main text.
2. A quantitative table of Peak width and expression in different conditions is needed.

Reviewer #4 (Remarks to the Author):

Summary:

Dai and colleagues explore the epigenetic changes that occur during methionine restriction (MR) in human cancer cells and mouse liver tissue. These results flesh out similar analyses that were part of the research group's 2015 Cell Metabolism publication, which already demonstrated global decrease in H3K4me3 in the context of MR for human HCT116 cancer cell lines and mouse liver.

As with their previous study, the authors report MR-associated decrease in global H3K4me3 levels (area, width) without changes in peak location, peak number, or set of genes affected. Examining gene and gene-set level changes in peak width, area, and height, they report significant pathway level associations with MR and peak width. Confusingly, these associations occur with pathways whose peak width <changes> in MR HCT116 and with genes whose peak width is <stable> in MR mouse liver, and not vice versa. Interestingly, peaks with sensitive width in MR HCT116 and peaks with robust width in MR mouse liver demonstrate significant enrichment in (different sets of) TF motifs. The authors also report correlations between peak dynamics (changes in height, area, width) and cis gene expression.

Assessment:

The data, findings, and approach are interesting. A primary concern is how the data and findings differ from the 2015 Cell Metabolism paper. If the findings are different – why? In addition, some of the claims in the main text are oversold and do not appear to be reflected in the data presented in the figures. In particular, the pathway level changes and peak width dynamics are mild and not straightforward to interpret. In addition, certain statements which could be easily substantiated through statistical hypothesis testing are not evaluated in such a manner. Finally, many of the “positive results” are oversold, especially in the text relative to what is shown in the figures. Overall, the gene and pathway level associations are not as strong as one might expect given the 2015 paper findings. Could that make this paper an interesting <negative result>, especially given the exhaustive peak geometry analysis that the authors have undertaken?

Specific critiques:

- Not clear how this paper relates to 2015 Cell Metabolism paper – including which of the data is novel. Which of the results are different (for example the 2015 paper reports CRC genes showing significant differences in peak intensity with MR). Are the conclusions different – if so how and why? i.e. is it because the data are different (different conditions, different assay) or are there analytic differences?
- Develop the negative results – is this what the authors expected when analyzing these data at

the peak or gene level? i.e. did they expect that MR would have a dramatic effect on a few key genes, or at least a biologically significant but global effect across the entire transcriptome. The authors are striving to derive biological meaning in the differential / non-differential peak distribution – what if there is none? Even if an effect exists, isn't it less significant than they expected. Assuming that their experiments are powered (ie they are inducing true MR, measuring H3K4me3 levels sensitively), then they should be able to at least comment on this, if not explicitly address with an analysis.

- The key analyses of the paper need to be substantiated with hypothesis testing, rather than the use of ad-hoc non-statistical reasoning. In addition, statistical hypotheses testing needs to be correctly applied in other cases.

- The key results of the paper are shown in Fig 2D and E lack p values – are these changes statistically significant? If so, what is the test that is applied? This analysis is essentially examining whether “more gene sets are significant than expected by chance” in condition 1 vs condition 2. One approach to do this analysis rigorously would be to compare the distribution of p values obtained by GSEA across all gene sets in condition 1 vs condition 2 using a Kolmogorov-Smirnoff test, or comparing each of these distributions vs uniform (ie expected p value distribution under the null). Since this is the central result of the paper – ie claiming that the set of genes with dynamic (or conserved) H3K4me3 peak width following MR are biologically important, this analysis needs to be done rigorously.

- Oddly, the q values shown in Fig 2D-E appear to be extremely significant for all conditions (average $-\log_{10} q$ value around 10 which corresponds to an average q value of 10^{-10} across these gene sets). This suggests that every gene set is significant in every analysis. How is this possible – was every gene set included in this plot? How were the gene set q values obtained (the methods show a very detail-poor description of pathway enrichment analysis which only reveals that “pathway enrichment analysis was conducted using msigdb”). More details need to be provided here, since GSEA is usually performed on gene expression data – how was this method adapted to ChIP-seq peaks? Furthermore, were only “significant” gene sets put into these figures rather than all the gene sets. That would also seem inappropriate, and still would not yield $-\log_{10} q$ values that are on average near ~ 10 . There is something wrong here, and requires either re-analysis or serious clarification.

- Ideally a gene set analysis method that takes into account confounders (e.g. a la limma) would be appropriate here. Especially since the authors notice that lost or gained peaks tend to be smaller (Fig S2D-E). This may be just a trivial result of differential peak analysis – since it may be hard to gain or lose large peaks (and instead you might just see a quantitative difference in their height or area). But if it is the case, then maybe the gene set results are just a function of smaller peaks being more likely to be differential – which in HCT116 those small peaks may cluster in (cancer specific pathways) and in liver maybe large peaks cluster in (liver) specific pathways. Can the authors correct for this effect? This also applies to the TF binding findings – could it be that small (ie “dynamic”) peaks in HCT116 and large (ie “robust”) peaks in liver are enriched in the reported motifs?

- Fig 4D and E also are presented without statistics or error bars, though visually there does not seem to be a significant difference in expression of “sensitive width genes” in HCT H3K4me3 (or similarly on robust width genes for mouse liver), even though these are highlighted. The legend does not provide the meaning of the square highlight.

- Certain statements in the text do not appear to be substantiated by the data

o Lines 208-210 refer to Fig 4J-K and report that “only H3K5me3 peak width dynamics significantly correlated with alteration in gene expression levels in both models”. However there appears to be significant (ie $P < 0.05$) correlation of both area and width in both models, and with height in mouse

liver in Fig 4J and K.

o Line 70-75: "We found aspects... expression" these two sentences are very unclear and appear to be run-on. They seem to strive to communicate the key points of the paper, but are almost incomprehensible and (more importantly) appear to be overselling the findings.

♣ Clarity: "The dynamics and consistency of the width, depending on context" I think refers to the fact that the peaks whose width <varied with MR> (aka "dynamics") in human cancer cell lines while the peaks whose width <did not vary with MR> (aka "consistency") in mouse liver seemed to track with pathway changes (aka "depending on context"). This is a very cryptic way to convey this point.

♣ Line 71: "were overall compressed" ... compression has a distinct meaning in the bioinformatics community eg with respect to information theory, please consider better word choice here to convey that the peaks width is smaller.

♣ Line 74: The final sentence dramatically oversells the correlation between peak width and gene expression: "... encoded nearly all features of gene activity including the physiological and pathophysiological program and the dynamics of gene expression". "Encoding nearly all features" implies that you could predict the gene expression profile exclusively from the peak width, which is certainly not demonstrated (or even tested). With that said – this would be an interesting analysis to pursue

We thank each reviewer for their constructive comments on manuscript. Modifications to the text are shown in **yellow** font. When additional analysis was conducted, the data are presented both in the revised manuscript and noted and also in response figures shown below. Thank you again for the constructive feedback that has served to improve our manuscript.

Reviewer #1 (Remarks to the Author):

In this manuscript, Dai et al. studied how nutrient availability influences the genomic architecture of H3K4me3 and gene expression and function. Analysis of genome-wide distribution of H3K4me3 in human cancer cells and mouse liver in response to alternations in methionine availability revealed that H3K4me4 peak width dynamics encodes biological information including transcription factor binding, gene expression and function. This manuscript addresses a timely and interesting question and makes some interesting observations. This work will be more impactful if experimental validation of observations/predictions can be done and a link between H3K4me3 width encoded transcription factor binding and gene expression can be established.

We thank the reviewer for the positive evaluation and helpful suggestions.

1. Figure 1. How does methionine restriction influence cancer cell growth and survival?

We thank the reviewer for raising this point. In our previous study (Cell Metabolism 2015, PMID: 26411344), we have measured growth curves for cells cultured under different concentrations of methionine and have shown that cell proliferation was reduced at longer times under methionine restriction (MR). Over time, in the methionine-restricted culture conditions we used, cell proliferation is reduced after many days but cell number is unaffected at the time we considered and the cells don't exhibit major stress responses such as p53 activation and senescence.

2. Figure 3. Can the authors validate the binding of some putative transcription factors in human cancer cells and mouse liver?

We thank the reviewer for this helpful advice. We have performed further analysis of the binding of some of the transcription factors by considering currently available public data which contains ChIP-seq profiles of several of the factors we identified in the revised manuscript. Specifically, we analyzed the binding of two transcription factors putatively associated with H3K4me3 width in each system (MYC and ELF1 in human cancer cells and HNF4a and RXRa in mouse liver, Response Figure 1A). We also included analysis of binding for CTCF in human cancer cells and E2F4 in mouse liver as negative controls (Response Figure 1B). Consistent with the TF-binding motif analysis, H3K4me3 peak subsets with sensitive width in human cancer cells and robust width in mouse liver were indeed enriched with binding-sites of these transcription factors (Response Figure 1C-H) but not with those as negative control (Response Figure 1I,J). We have included these

analysis in the revised manuscript (Figure S9). We have also included a description of the methods used in this additional analysis in the Methods as well.

3. Figure 4. As histone modifications need to orchestrate specific transcription factor binding to regulate transcription, can the authors perform an integrative analysis of H3K4me3 dynamics, cell-type specific transcription factor binding and gene expression?

We thank the reviewer for this comment. We have now performed additional integrative analysis of H3K4me3 dynamics, cell-type specific transcription binding and gene expression. We have compared both H3K4me3 dynamics and its relationship with differential gene expression in peaks bound by different number of cell-type specific transcription factors (Response Figure 2). We found that binding of cell-type specific transcription factors significantly correlated with H3K4me3 dynamics (especially width dynamics) and observed stronger correlation between H3K4me3 width changes and gene expression changes in peaks bound by more TFs in both systems. We have included this analysis in the revised manuscript (Figures S10, S13G, and S13H).

Reviewer #2 (Remarks to the Author):

The manuscript by Dai et al is focused on the relevance of H3K4me3 peak width dynamics in regulating gene expression and, consequently, biological functions. The authors used the methionine restriction (MR) as experimental paradigm in vitro (HCT116 cells) and in vivo (liver of C57BL6 mice exposed to a control diet or to a low methionine diet). They investigated the genome-wide changes in H3K4me3, analyzing height, area and width of the ChIP-seq peaks, and concluded that peak width directly correlates with gene expression (RNA-seq) changes and stress that this parameter is linked to key biological process (acquisition of cell identity) and pathophysiological mechanisms (tumor suppression and progression).

Major comments

- The aim of the manuscript is to identify a new relevant parameter of an epigenomic analysis (the width of H3K4me3 peak) to study the regulation of gene expression and, lastly, the modulation of biological functions. To this end, the authors used an experimental paradigm of methionine restriction. As previously published by the same group (Histone Methylation Dynamics and Gene Regulation Occur through the Sensing of One-Carbon Metabolism, Cell Metab. 2015 Nov 3; 22(5): 861–873, Mentch et al.) the methionine restriction in HCT116 cells affect the global H3K4me3 profile. Similar to their previously published results, they

showed an overall reduction in H3K4me3 in peak intensity (Figure S1E, S1F). However they did not detect any difference in the peak number between cancer cells exposed to 100uM or 3uM methionine (Figure 1B), indicating that methionine restrictions has no effect on genomic distribution of this histone modification. Thus, they decided to focus on the peak descriptors and they identify peak width as a relevant feature correlating with gene activation. Several times in the manuscript the authors reported that the peak width of ChIP seq analysis could be correlated to activation/inactivation of specific biological functions. However, based on the results they provide, it is not possible to establish a direct correlation between peak width and biological processes (except for a certain gene expression profile). To make their observation more relevant, the authors should modify the experimental paradigm. Since the authors chose a cellular model of cancer, they should demonstrate how changes in H3K4me3 peak width influences cancer cells in terms of proliferation/function, as a proof of principle that the alteration of this parameter in an epigenetic analysis really correlates with a functional output.

We thank the reviewer for raising this concern. We completely agree that the effects on cell proliferation are important factors to consider. In our previous study (Cell Metabolism 2015, PMID: 26411344), we have measured cell proliferation under different concentrations of methionine and have shown that cell proliferation was reduced under MR (Figure 1H in Cell Metabolism 2015, PMID:26411344) with no gross effects on stress and cell survival at the time point we considered.

To provide a more comprehensive assessment of MR and H3K4me3 dynamics in relation to cell proliferation and survival, we further considered gene expression associated with proliferation and survival related genes with the RNA-seq data we collected. We also integrated our findings on H3K4me3 dynamics with related compendia on required genes. We have compared H3K4me3 and gene expression changes in proliferation and survival related genes (i.e. fitness genes) identified in a CRISPR-based screening (Cell 2015, PMID: 26627737) to all other genes and now show that fitness genes exhibit more dramatic H3K4me3 peak width changes and are more likely to be down-regulated in response to MR (Response Figure 3). We have included these results in the revised manuscript (Figure S15).

- To analyze the effect of long-term alterations in nutrient availability, the authors used an in vivo model, focusing on the liver. However, the rationale of the choice of this paradigm is not properly justified in the manuscript. To strengthen the take-home-message the authors should perform further experiments in an in vivo model of cancer, investigating the effect (epigenetic and functional) of methionine restriction in this model. If there are scientific reasons making this experiment not feasible, the authors should clearly justify them, and further explain why they chose to study the effect of MR in the liver.

We thank the reviewer for raising this concern and apologize for the confusion. We completely agree that an in vivo tumor model will be a good extension of our current work. We chose to use healthy mice as the in vivo model to increase the diversity of model systems by covering both short- and long-term, both in vitro and in vivo, and both pathologic and healthy conditions. We believe such choice of a broad spectrum of models will help to unravel general principles. Liver was used in profiling the epigenetics and transcriptomics mainly because it is the metabolic organ which is most responsive to metabolic reprogramming and there are liver-related phenotypes associated with MR related to metabolic health. We have added clarification of this point in the revised manuscript.

- Results reported in figure 2 highlight that relevance of MR on peak width is different in cancer cells and in liver: the GSEA analysis showed that in HCT116 cells the change of peak width (sensitive peaks) exhibits significant enrichment of specific processes. On the other hand, in the liver the correlation is between the consistency of the peak width (robust peaks) with biological information. Which is the meaning of this result? The authors picked the liver model to investigate the long-term effect of MR and to strengthen the in vitro observations. However, the outcome is different: robust peaks in the liver are the ones correlating with

enrichment of biological processes. Does this mean that under MR a set of genomic regions (that are the ones related to relevant biological processes, such as metabolism) in the liver show conserved peak width of H3K4me3? If so:

We appreciate this thoughtful comment. We suspect it reflects the difference between tumor and normal tissue. Cell-type specific functions in normal tissues are more robust under alterations in environmental variables to ensure normal functioning of this organ, while cancer cells have more flexibility to maximize fitness in a varying environment. We understand that this explanation is only a hypothesis on this dramatic difference in epigenomic dynamics between cancer cells and normal tissue and its validation is far from straightforward, but these differences of robustness and sensitivity in the two systems do define a general principle. The principle is that the response of the width is the most informative under MR we have defined in both models. We have added further discussion of this point in the revised text.

1) Is this related to the experimental paradigm used (for example exposure to the 0.84% or 0.12% methionine diet was too short)? According to this hypothesis, the number of genes modulated by MR in the liver (in terms of gene expression, figure 4G) is lower as compared to the number of differentially regulated genes in cancer cells. The use of a different in vivo model could help in resolving this discrepancy.

We thank the reviewer for raising this concern. In previous work, we have quantified bulk levels of H3K4me3 in liver and observed significant differences between mice fed with control diet and methionine-restricted diets (Figures 5E,F in Cell Metabolism 2015, PMID:26411344). We believe such differences in bulk levels of H3K4me3 provide rationale for using this diet. There is also substantial literature showing that while methionine deprivation is toxic to animals, this defined diet is well tolerated and produces health benefits (e.g. Metabolism 2013, PMID: 23928105; PLoS One 2012, PMID: 23236485; Sci Rep 2015, PMID: 25744495). We have further emphasized this rationale in the revised manuscript. Second, we note that differential gene expression in vivo is confounded by more factors including composition of liver by different cell types and variation in gene expression between individual mice, which result in larger within-group variation and less significant differences. We have included discussion related to this point in the revised manuscript.

2) Or does it mean that differences of peak width on genes related to metabolism in liver is less dynamic compared to cancer cells and consequently more conserved in response to external stimuli, such as MR?

The reviewer is correct. As our analysis suggests, peak width on genes related to metabolism in liver (which was considered as liver-specific function) is less dynamic compared to peak width on genes related to proliferation and cancer in human cancer cells at least in response to MR.

- This manuscript suggests a new interesting parameter that can give insights in analyzing ChIP-seq data. Do the authors think that peak width could be more relevant than the

H3K4me3 enrichment itself? Do they compare the correlation between H3K4me3 and gene expression and the correlation between H3K4me3-peak width with gene expression? Do the author think this can be relevant and potentially explain why mild changes in H3K4me3 enrichment can correlate with big changes in gene expression?

We thank the reviewer for the positive comment and for raising these questions.

We believe peak location and peak area are encoding different aspects of association between H3K4me3 and gene expression. For example, enrichment of H3K4me3 around transcription start sites is associated with active transcription (Nature 2002, PMID: 12353038), while broad H3K4me3 peaks are associated with cell-type specific genes and reduced variation in gene expression (Cell 2014, PMID: 25083876 and Nature Genetics 2015, PMID: 26301496).

In our study, we also correlated all three H3K4me3 peak descriptors with gene expression levels and found significant positive correlation between all three peak descriptors and gene expression, which is consistent with the known relationship between H3K4me3 enrichment and active gene expression (Figure 4B, 4C). Nevertheless, regarding the dynamics, the stronger signal is contained in the peak width.

Our study shows that peak width appears to contain the most dynamic information of H3K4me3 at least in response to nutrient availability. While the genomic location is preserved the response does contain larger changes in the overall profile such as the peak compression.

We have included clarification of these points in the revised manuscript.

Minor comments

- The authors should provide further information about the statistical analysis that has been performed. If they mention statistically significant changes they should indicate the p values in the graph and in the figure legends they should give information about the type of statistical test they used.

We thank the reviewer for this suggestion. We have included more details and more statistical analysis in the revised manuscript both in the text and figures.

- I recommend a revision of the manuscript by an English native speaker.

We have further edited the manuscript before resubmission.

Reviewer #3 (Remarks to the Author):

Dai et al. performed a systematic analysis of H3K4me3 peak shape and found strong associations between gene expression/pathway enrichment/TFBSs and peak width, but not

peak height or peak area. In general, the observation is interesting and the conclusions are well supported by the data. There are some issues that need further clarification.

We are appreciative of the positive remarks. We are happy to include further clarification in the revised manuscript.

Major:

1. HCT116 cells are treated by methionine with two different concentrations. But the authors do not find much difference under the two conditions in Fig 1. How did the authors choose such two concentrations and is the 3uM sufficient to alter chromatin?

We thank the reviewer for raising this concern. Methionine concentrations in the control and methionine-restricted medium were chosen based on our previously published study (Cell Metabolism 2015, PMID: 26411344) in which we observed significant alterations in methionine metabolism and reduction in H3K4me3 bulk levels at 3uM methionine (Figure 1F,G in Cell Metabolism 2015, PMID: 26411344). Thus, we previous published work has shown that 3uM methionine in the medium is sufficient to induce large changes in the bulk levels of histone methylation particularly at H3K4me3. We have added clarification of this point to the revised manuscript.

3. Sensitive width in human data is enriched in GSEA analysis, whereas Robust Width in mouse data is enriched (Fig 2 ~ 4). The authors need to explain in detail why there is such dramatic discrepancy in two closely related species, and need to rule out the possibilities of any technical issues.

We agree that our observations in cancer cells and normal liver are interesting. We have added further discussion of these differences in the revised manuscript. We also completely agree that addressing all technical possibilities is essential.

We understand that analysis of ChIP-seq data may be affected by technical issues such as peak calling and data normalization methods and have investigated many of the possibilities regarding these effects to the best of our ability by using alternative data analysis methods in the current manuscript (Figure S1A-C and Figure S7F in the previous version of manuscript for human cancer cells).

We have further expanded the comparative analysis of alternative data analysis approaches to include more features and additional analysis of the ChIP-seq data from mouse liver as well in the revised manuscript. In the peak calling methods tested, we have compared multiple outputs including number of peaks (Response Figure 4A,B), genomic regions identified as peaks (Response Figure 4C,D), number of reads in peaks (Response Figure 4E,F) and genes marked by peaks (Response Figure 4G,H). We also evaluated the consistency of peak height, area and, width changes evaluated based on raw and extended read coverage profiles (Response Figure 5A,B) and under different peak-calling pipelines (Response Figure 5C,D). Taken together, these results suggest that quantification of H3K4me3 peak dynamics was highly consistent under different ChIP-seq data analysis pipelines. We have included these results in the revised manuscript (Figure S3, S4).

We believe that sensitive width in human cancer cells and robust width in mouse liver may reflect the differences between cancerous and healthy tissues in responding to alteration in environmental factors. Cell-type-specific functions in normal tissues are more robust under alteration in environmental variables to ensure normal functioning of this organ, while cancer cells have more flexibility to maximize fitness in a varying environment. We understand that this explanation is only a hypothesis but the principle that the peak width is the most informative parameter governing H3K4me3 dynamics upon changes in nutrient availability is conserved in both models. We have added further discussion of this point in the revised text.

Response Figure 5 (now Figure S4) (A-B) Spearman correlation between raw (i.e. without extension of reads to the whole fragment) and extended (i.e. with extension of reads to the whole fragment) read coverage profiles for changes in height, area and width in (A) human cancer cells and (B) mouse liver. (C-D) Spearman correlation between peak calling algorithms for changes in height, area and width in (C) human cancer cells and (D) mouse liver.

4. in Fig 3f, the association of sensitive width with TSG or robust width with OG is not convincing enough. A more rigorous statistical analysis is needed.

We thank the reviewer for raising this concern. We have now performed a statistical analysis using a chi-squared test to assess significance of the enrichment of TSG in robust width and OG in sensitive width ($p=0.0324$). We have included this in the revised manuscript.

Minor:

1. Fig 4C is not mentioned in the main text.

We apologize for this and thank the reviewer for pointing it out. We have corrected this typo in the revised manuscript.

2. A quantitative table of Peak width and expression in different conditions is needed.

We thank the reviewer for this comment and completely agree. We have added this to the revised manuscript as Table S1.

Reviewer #4 (Remarks to the Author):

Summary:

Dai and colleagues explore the epigenetic changes that occur during methionine restriction (MR) in human cancer cells and mouse liver tissue. These results flesh out similar analyses that were part of the research group's 2015 Cell Metabolism publication, which already demonstrated global decrease in H3K4me3 in the context of MR for human HCT116 cancer cell lines and mouse liver.

As with their previous study, the authors report MR-associated decrease in global H3K4me3 levels (area, width) without changes in peak location, peak number, or set of genes affected. Examining gene and gene-set level changes in peak width, area, and height, they report significant pathway level associations with MR and peak width. Confusingly, these associations occur with pathways whose peak width <changes> in MR HCT116 and with genes whose peak width is <stable> in MR mouse liver, and not vice versa. Interestingly, peaks with sensitive width in MR HCT116 and peaks with robust width in MR mouse liver demonstrate significant enrichment in (different sets of) TF motifs. The authors also report correlations between peak dynamics (changes in height, area, width) and cis gene expression.

Assessment:

The data, findings, and approach are interesting. A primary concern is how the data and findings differ from the 2015 Cell Metabolism paper. If the findings are different – why? In addition, some of the claims in the main text are oversold and do not appear to be reflected in the data presented in the figures. In particular, the pathway level changes and peak width dynamics are mild and not straightforward to interpret. In addition, certain statements which could be easily substantiated through statistical hypothesis testing are not evaluated in such a manner. Finally, many of the “positive results” are oversold, especially in the text relative to what is shown in the figures. Overall, the gene and pathway level associations are not as strong as one might expect given the 2015 paper findings. Could that make this paper an interesting <negative result>, especially given the exhaustive peak geometry analysis that the authors have undertaken?

We thank the reviewer for the positive remarks and helpful comments. We have added further clarification of how this study extends from previous work that focused on bulk levels of H3K4me3 and a cursory qualitative analysis of one ChIP-seq experiment. We have also worked to temper statements where appropriate, add further statistical analysis and provide additional discussion to improve the clarity. Detailed, point-by-point responses are below.

Specific critiques:

- Not clear how this paper relates to 2015 Cell Metabolism paper – including which of the data is novel. Which of the results are different (for example the 2015 paper reports CRC

genes showing significant differences in peak intensity with MR). Are the conclusions different – if so how and why? i.e. is it because the data are different (different conditions, different assay) or are there analytic differences?

We thank the reviewer for raising the question. While the main advance of the previous paper was to show that methionine availability modulates histone methylation bulk levels through influencing the methionine cycle, this study sought to characterize the dynamics of the genomic architecture of H3K4me3 under MR and quantify its connection with alterations in gene expression. We applied peak geometry based analysis and performed additional ChIP-seq and RNA-seq analysis in this study. With these advances we were able to uncover a principle highlighting importance of H3K4me3 peak width dynamics at least in the context of MR. We have added clarification of these points and place this current study in additional context as it relates to the 2015 Cell Metabolism paper.

- Develop the negative results – is this what the authors expected when analyzing these data at the peak or gene level? i.e. did they expect that MR would have a dramatic effect on a few key genes, or at least a biologically significant but global effect across the entire transcriptome. The authors are striving to derive biological meaning in the differential / non-differential peak distribution – what if there is none? Even if an effect exists, isn't it less significant than they expected. Assuming that their experiments are powered (ie they are inducing true MR, measuring H3K4me3 levels sensitively), then they should be able to at least comment on this, if not explicitly address with an analysis.

We thank the reviewer for raising this concern. We agree that it is an important yet unanswered question regarding whether MR changes histone methylation globally or locally, which is also the main question that we sought to answer in this study. First, we believe that the MR experiments we performed in this study were sufficient to change the chromatin based on the quantification of histone methylation bulk levels in our previous Cell Metabolism paper (Figure 1C and Figure 5E in Cell Metabolism 2015, PMID:26411344). Nevertheless, the evidence we provided in this study points to the informative nature of width dynamics compared to alterations in other peak descriptors, which we believe to be clarifying at least two points: (1) there is indeed specificity in the epigenetic response to MR; (2) width dynamics is the most informative variable in how MR influences histone methylation and gene expression.

We have provided additional clarification related to these points in the revised manuscript.

- The key analyses of the paper need to be substantiated with hypothesis testing, rather than the use of ad-hoc non-statistical reasoning. In addition, statistical hypotheses testing needs to be correctly applied in other cases.

We thank the reviewer for pointing to this. We have included more details about quantitation and statistical analysis and have conducted additional statistical analysis in the revised manuscript. This is shown both in the text and in the figures through the manuscript.

- The key results of the paper are shown in Fig 2D and E lack p values – are these changes statistically significant? If so, what is the test that is applied? This analysis is essentially examining whether “more gene sets are significant than expected by chance” in condition 1 vs condition 2. One approach to do this analysis rigorously would be to compare the distribution of p values obtained by GSEA across all gene sets in condition 1 vs condition 2 using a Kolmogorov-Smirnoff test, or comparing each of these distributions vs uniform (ie expected p value distribution under the null). Since this is the central result of the paper – ie claiming that the set of genes with dynamic (or conserved) H3K4me3 peak width following MR are biologically important, this analysis needs to be done rigorously.

We thank the reviewer for raising this concern and agree that more statistical analyses will improve the manuscript. For Figs 2D and 2E, we have performed additional hypothesis testing including Wilcoxon rank-sum test and Kolmogorov-Smirnov test to compare Q-value distributions in different H3K4me3 peak subsets and visualized the comparison by Quantile-Quantile (Q-Q) plots in the revised manuscript (Response Figure 6A, 6B). We have also performed such statistical analysis for distributions of TF binding motif enrichment Q-values (Response Figure 6C, 6D). These new results are included in the revised manuscript (Figure S5, S8A, S8B).

Response Figure 6 (now Figures S5, S8A, S8B) (A-B) Quantile-Quantile (Q-Q) plots, Wilcoxon rank-sum P-values and Kolmogorov-Smirnov P-values comparing distributions of pathway enrichment Q-values in other peak subsets to the peak subset with (A) sensitive width in human cancer cells and (B) robust width in mouse liver. (C-D) Q-Q plots, Wilcoxon rank-sum P-values and Kolmogorov-Smirnov P-values comparing distributions of TF binding motif enrichment Q-values in other peak subsets to the peak subset with (C) sensitive width in human cancer cells and (D) robust width in mouse liver.

• Oddly, the q values shown in Fig 2D-E appear to be extremely significant for all conditions (average $-\log_{10}$ q value around 10 which corresponds to an average q value of 10^{-10} across these gene sets). This suggests that every gene set is significant in every analysis. How is this possible – was every gene set included in this plot? How were the gene set q values obtained (the methods show a very detail-poor description of pathway enrichment analysis which only reveals that “pathway enrichment analysis was conducted using msigdb”). More details need to be provided here, since GSEA is usually performed on gene expression data – how was this method adapted to ChIP-seq peaks? Furthermore, were only “significant” gene sets put into these figures rather than all the gene sets. That would also seem inappropriate, and still would not yield $-\log_{10}$ q values that are on average near ~ 10 . There is something wrong here, and requires either re-analysis or serious clarification.

We thank the reviewer for raising this concern and apologize for the confusion. The significant enrichment of pathways in all peak sets is the direct consequence of pathway enrichment in genes marked by H3K4me3. All gene sets we considered in Fig 2D-E are indeed subsets of this set, thus a gene set enriched in H3K4me3 genes will also be enriched in these subsets. Thus, we used a bootstrapping-based approach, in which we generated

three random subsets of genes with H3K4me3 peaks and used enrichment Q-value distributions in these three random H3K4me3 gene sets as a null distribution.

In the revision, we have implemented the GSEA algorithm using all genes with H3K4me3 as the reference set to provide an additional analysis that corrects for the background H3K4me3 gene set and now show that the enrichment of cell type specific functions in peaks with sensitive width in human cancer cells and peaks with robust width in mouse liver was retained (Response Figure 7). This additional analysis is included in the revised manuscript (Figure S7). We have also included more a detailed description of our modification to the GSEA algorithm in the Methods of the revised manuscript.

Response Figure 7 (now Figure S7) (A-B) Distributions of pathway enrichment Q-values estimated using all genes with H3K4me3 as the reference set in (A) human cancer cells and (B) mouse liver. P-values were computed from Wilcoxon rank-sum test. (C-D) Number of significantly enriched pathways in peak subsets in (C) human cancer cells and (D) mouse liver. (E-F) Annotation of significantly enriched pathways in peak subset with (E) sensitive width in human cancer cells and (F) robust width in mouse liver. (G-H) Q-Q plots, Wilcoxon rank-sum P-values and Kolmogorov-Smirnov P-values comparing distributions of pathway enrichment Q-values in other peak subsets to the peak subset with (G) sensitive width in human cancer cells and (H) robust width in mouse liver.

- Ideally a gene set analysis method that takes into account confounders (e.g. a la limma) would be appropriate here. Especially since the authors notice that lost or gained peaks tend

to be smaller (Fig S2D-E). This may be just a trivial result of differential peak analysis – since it may be hard to gain or lose large peaks (and instead you might just see a quantitative difference in their height or area). But if it is the case, then maybe the gene set results are just a function of smaller peaks being more likely to be differential – which in HCT116 those small peaks may cluster in (cancer specific pathways) and in liver maybe large peaks cluster in (liver) specific pathways. Can the authors correct for this effect? This also applies to the TF binding findings – could it be that small (ie “dynamic”) peaks in HCT116 and large (ie “robust”) peaks in liver are enriched in the reported motifs?

We thank the reviewer for this insightful comment and apologize for the confusion around the definition of robust and sensitive peaks. The reviewer is correct about the tendency of gained or lost peaks to be smaller. This is because for smaller peaks, their H3K4me3 enrichment is closer to the threshold used in peak-calling, thus potentially resulting in their absence in one condition. To avoid this bias, the method we used to define the robust or sensitive peaks was based on fold changes of height, area or width in MR condition compared to the control instead of gain or loss under MR (Figure 2C in the manuscript). In addition, peak regions in all samples were merged to include smaller peaks called only in one condition. It is also notable that the enrichment of pathways and TFs was only observed in sensitive width (neither sensitive height nor sensitive area) in cancer cells and robust width (neither robust height nor robust area) in mouse liver, which supports the importance of width dynamics compared to that of area and height. We have added clarification of this point in the revision.

• Fig 4D and E also are presented without statistics or error bars, though visually there does not seem to be a significant difference in expression of “sensitive width genes” in HCT H3K4me3 (or similarly on robust width genes for mouse liver), even though these are highlighted. The legend does not provide the meaning of the square highlight.

We thank the reviewer for raising this concern and apologize for this omission. We have included additional hypothesis testing including Wilcoxon rank-sum test and Kolmogorov-Smirnov test to compare the distribution of gene expression levels in these peak subsets in the revised manuscript (Response Figure 8). The P-values from these tests indicate that the higher expression levels of sensitive width genes in human cancer cells and that of robust width genes in mouse liver are significant. These results are included in the revised

Response Figure 8 (now Figure S13) (A-B) Q-Q plots, Wilcoxon rank-sum P-values and Kolmogorov-Smirnov P-values comparing distributions of gene expression levels associated with other peak subsets to the peak subset with (A) sensitive width in human cancer cells and (B) robust width in mouse liver.

manuscript (Figure S13). The squares in these figures are highlighting sensitive width in human cancer cells and robust width in mouse liver, which have been demonstrated to associate with cell-type-specific biological functions and TF binding in previous analyses. We have modified the legend to clarify this point in the revised manuscript.

- Certain statements in the text do not appear to be substantiated by the data

We thank the reviewer for pointing out this. We have conducted additional textual edits especially regarding statistical analysis used to make conclusions.

o Lines 208-210 refer to Fig 4J-K and report that “only H3K5me3 peak width dynamics significantly correlated with alteration in gene expression levels in both models”. However there appears to be significant (ie $P < 0.05$) correlation of both area and width in both models, and with height in mouse liver in Fig 4J and K.

We thank the reviewer for this comment. We have modified as appropriate this statement in the revised manuscript.

o Line 70-75: “We found aspects... expression” these two sentences are very unclear and appear to be run-on. They seem to strive to communicate the key points of the paper, but are almost incomprehensible and (more importantly) appear to be overselling the findings.

We thank the reviewer for this comment. We have made textual changes in the revised manuscript to improve the clarity and provide further justification.

♣ Clarity: “The dynamics and consistency of the width, depending on context” I think refers to the fact that the peaks whose width <varied with MR> (aka “dynamics”) in human cancer cell lines while the peaks whose width <did not vary with MR> (aka “consistency”) in mouse liver seemed to track with pathway changes (aka “depending on context”). This is a very cryptic way to convey this point.

We thank the reviewer for noting this. The reviewer is correct on the exact meaning of this statement. We have modified this statement to improve clarity in the revised manuscript.

♣ Line 71: “were overall compressed” ... compression has a distinct meaning in the bioinformatics community eg with respect to information theory, please consider better word choice here to convey that the peaks width is smaller.

We thank the reviewer for this comment. We have changed ‘compressed’ to ‘reduced’ to avoid confusion.

♣ Line 74: The final sentence dramatically oversells the correlation between peak width and gene expression: “... encoded nearly all features of gene activity including the physiological and pathophysiological program and the dynamics of gene expression”. “Encoding nearly all features” implies that you could predict the gene expression profile exclusively from the peak

width, which is certainly not demonstrated (or even tested). With that said – this would be an interesting analysis to pursue

We thank the reviewer for raising this point. Here by ‘encoding nearly all features’ we are referring to the association of peak width dynamics with cell-identity-related functions and gene expression alterations, which is not observed for height and area dynamics. We apologize for the confusion. We have modified this statement to improve clarity in the revised manuscript.

Reviewers' comments:

Reviewer #1 (Remarks to the Author):

The authors have successfully addressed my previous concerns. The manuscript is ready for publication.

Reviewer #2 (Remarks to the Author):

The revised version of the manuscript entitled "Methionine metabolism influences the genomic architecture of H3K4me3 with the link to gene expression encoded in peak width", looks improved and I believe the authors made a good job in addressing the raised concerns.

To reply to the author's response to point 2 of my concern ("We thank the reviewer for raising this concern and apologize for the confusion. We completely agree that an in vivo tumor model will be a good extension of our current work. We chose to use healthy mice as the in vivo model

We have added clarification of this point in the revised manuscript"), I understand that repeating similar studies in a model of cancer will be a massive amount of work and not strictly necessary in elucidation of the take home message of this manuscript.

I only have a suggestion regarding the sentence at page 3 of the manuscript. I noticed the authors added/modified the following sentence in the manuscript: "We considered a mouse model of dietary methionine restriction leading to changes in bulk levels of H3K4me3 in liver and cultured human cancer cells (HCT116) subjected to methionine restriction in culture media that also leads to changes in bulk levels of H3K4me3". I would recommend re-phrase it as follows: "We considered a mouse model of dietary methionine restriction and focused our analysis on liver. In this organ the chosen dietary regimen imposes changes in bulk levels of H3K4me3. Similar changes occur in cultured human cancer cells (HCT116) subjected to methionine restriction in culture media, making the in vivo model adopted in this study feasible to investigate the relevance of a MR-related epigenetic and transcriptional reprogramming to metabolic health".

Reviewer #3 (Remarks to the Author):

The authors have adequately addressed my concerns.

Reviewer #4 (Remarks to the Author):

The authors have implemented additional analyses and changed the text to address many of my concerns. The rebuttal is well organized, and clear. The overall conclusions are still perplexing, both with respect to the differences between cancer cell lines and mouse liver, and the "exceptionality" of peak width as a parameter. In particular, not clear how "robust width" peaks could drive MR biology in mouse liver if they are not associated with differentially expressed genes. In general, the exceptional role for width as an important parameter of peak geometry is not robustly supported by the gene expression analysis in Figure 4.

Specific concerns:

Fig 2

The authors still have not described what they mean by the "GSEA algorithm". Are they referring to the actual GSEA method - Subramanian PNAS which would require computing an enrichment score by applying a modified KS statistic to a distribution of r^2 values for all genes (ie significant

vs non-significant) and comparing this enrichment score to sample label permutations. If so, they should describe how they adapted this method to chip-seq analysis or supply a reference to a method that does.

My guess is that they associating the "top 500 peaks" with annotated MSigDb gene sets and getting gene set p values using a hypergeometric test. This is not GSEA but just a hypergeometric test and should be described as such in the methods. Also they should justify why they used "top 500" instead of a more standard threshold for choosing top peaks / genes (i.e. FDR, q value, bonferroni corrected P) as inputs to this kind of analysis.

The analysis in Fig S7 should be in the main text, to replace Figs 2D-E. From a hypothesis testing perspective, there are no differences between a set of significant hypotheses that have a median p value $1e-20$ vs a median p value / q value $1e-20$. The question you want to ask is whether "more gene sets are significantly perturbed" - if everything is significant it doesn't matter if one condition has q values $1e-20$. They are essentially identical. As the authors note, the "pan-significance" of gene sets in this figure has to do with the fact that most functional gene sets will be enriched in H3K4me3 peaks. Given this, the analysis shown in S7 is the correct way to do the pathway analysis, i.e. restricting the "universe" of gene sets to genes with H3K4me3 peaks. The pathways that are quoted in the main Figure should come from this, more correct, analysis.

I think Figs S7C and S7D are the best way to address the hypothesis that sensitive width / robust width show the highest degree of "biological perturbation". The results (that 70% of sensitive width sets and 40% of robust width sets relative to baseline of 0-10%) are more compelling than quantitative differences in p or q values. These should be shown in the main text, but with p values comparing pairs of bars in this plot (eg via fisher test or binomial test)). i.e. what matters more than a quantitative difference in Q values is that a significantly larger proportion of gene sets are significant (subject to a fixed, reasonable p / q value threshold) among the peak types of interest.

Fig 4 -

Panels B and C -

The use of bar charts is not clear here, especially to indicate extreme p values. The number of peaks that are included in such an analysis will make the weakest correlation appear very significant, and indeed here the correlation is not very strong ($R^2 = 0.09$). In addition, the reader is tempted to draw comparisons between pairs of bars, but no hypothesis test is being communicated. Presumably, the authors want to show that there is a significant but weak correlation between expression and H3Kme3 that is not sensitive to MR. (ie the correlation appears similar between red and blue bars in both cancer cell line and mouse liver). One hypothesis test would be to test for differential correlation between H3K4me3 peak size descriptors and gene expression across the two conditions, e.g. using a linear model or ANOVA with gene expression as a dependent variable, height and/or area and/or width as independent numeric variables and MR as a binary factor.

More importantly, the influence of peak width on gene expression seems to be the weakest of all three size descriptors. How does that play with the story developed by the authors - i.e. that peak width encodes important aspects of H3K4me3 biology? This important inconsistency does not seem to be addressed by the authors.

Panels D and E - It's not clear what the biological significance of steady state MR gene expression as a function of peak category. For sensitive width genes in human cancer, it would seem to make more sense to examine differential expression in MR vs normal relative, i.e. jump to 4J and/or even demonstrate a scatter or density plot of peak width vs gene expression showing the

correlation. As for robust width genes - what does it mean that they are the most highly expressed in MR Mouse Liver (among the peak categories investigated). Are these also highly expressed in normal mouse liver? If these robust width genes are not differentially expressed, then are they truly biologically significant or important in MR? Or is the complement of this set important (non robust width, which are presumably differentially expressed).

Panels J and K - The authors should not be bar plotting $-\log_{10} p$ values, since $-\log_{10} P$ magnitude differences have little meaning in this range. Moreover the plotting is visually confusing as one is trying to compare different bars - concealing the key patterns in the data. Furthermore, the authors should (like in 4B and 4C) perform hypothesis testing to ask which of these factors (height, area, width) is best predictive of gene expression (e.g. using a linear model) and providing p values on pairs of bars. How do those predictions play with their hypotheses from Figs 1-3. It seems, for example, in Panel J that Width would be best predictive of MR dependent gene expression changes in cancer cell lines, consistent with the authors' assertion that peak width changes are biologically important here.

Panels K - If mouse liver gene peak width changes significantly correlate with gene expression does that mean that genes in robust width peaks are not differentially expressed in MR? How do the authors reconcile this with their model.

Small critiques

- Line 161 "for each subset of peaks the distribution of the enrichment Q values of the top 100 enriched (one-sided Fisher's exact Q values < 0.05) pathways"

This does not make sense. Which of the two is it? The top 100 pathways or all pathways with $Q < 0.05$? If top 100 then why 100? A Q value threshold is more rigorous / standard. If both, then something is wrong in the analysis - since $Q < 0.05$ should give different numbers of pathways in each peak subset.

- This awkward sentence does not make sense ..

> Lines 70-72

> We found that aspects of the peak geometry such as the height and area were overall reduced
> that accounted for most of the global changes.

"nearly all aspects of H3K4me3 biology" is too broad of a claim, please tone down or clarify ... "all aspects of H3K4me3 biology" is a concept that is hard to qualify, but I imagine would include features that would not be captured by a ChIP-seq profiling study, for example kinetics, protein complex interactions, and three dimensional nuclear biology. This statement is even harder to justify when one considers some of the more perplexing conclusions of this work.

- Suggested replacement e.g. "reflected important cellular processes previously linked to H3K4me3 histone modifications"

> Lines 72-75

> "Strikingly, however, while the most conserved feature of H3K4me3 dynamics was found in the
> peak width, changes in peak width but not other features of peak geometry were associated
> with nearly all aspects of H3K4me3 biology including cell identity related gene expression
> programs and the dynamics of gene expression."

Response to Reviewers' comments:

Reviewer #1 (Remarks to the Author):

The authors have successfully addressed my previous concerns. The manuscript is ready for publication.

We thank the reviewer for the positive comment on our revisions.

Reviewer #2 (Remarks to the Author):

The revised version of the manuscript entitled "Methionine metabolism influences the genomic architecture of H3K4me3 with the link to gene expression encoded in peak width", looks improved and I believe the authors made a good job in addressing the raised concerns.

To reply to the author's response to point 2 of my concern ("We thank the reviewer for raising this concern and apologize for the confusion. We completely agree that an in vivo tumor model will be a good extension of our current work. We chose to use healthy mice as the in vivo model We have added clarification of this point in the revised manuscript"), I understand that repeating similar studies in a model of cancer will be a massive amount of work and not strictly necessary in elucidation of the take home message of this manuscript.

I only have a suggestion regarding the sentence at page 3 of the manuscript. I noticed the authors added/modified the following sentence in the manuscript: "We considered a mouse model of dietary methionine restriction leading to changes in bulk levels of H3K4me3 in liver and cultured human cancer cells (HCT116) subjected to methionine restriction in culture media that also leads to changes in bulk levels of H3K4me3". I would recommend re-phrase it as follows: "We considered a mouse model of dietary methionine restriction and focused our analysis on liver. In this organ the chosen dietary regimen imposes changes in bulk levels of H3K4me3. Similar changes occur in cultured human cancer cells (HCT116) subjected to methionine restriction in culture media, making the in vivo model adopted in this study feasible to investigate the relevance of a MR-related epigenetic and transcriptional reprogramming to metabolic health".

We thank the reviewer for the positive remarks. We greatly appreciate the helpful advice and have re-phrased this sentence in the revised manuscript.

Reviewer #3 (Remarks to the Author):

The authors have adequately addressed my concerns.

We thank the reviewer for positive evaluation of the revisions.

Reviewer #4 (Remarks to the Author):

The authors have implemented additional analyses and changed the text to address many of my concerns. The rebuttal is well organized, and clear. The overall conclusions are still perplexing, both with respect to the differences between cancer cell lines and mouse liver, and the "exceptionality" of peak width as a parameter. In particular, not clear how "robust width" peaks could drive MR biology in mouse liver if they are not associated with differentially expressed genes. In general, the exceptional role for width as an important parameter of peak geometry is not robustly supported by the gene expression analysis in Figure 4.

We thank the reviewer for positive comments on the revisions. Yes, we did spend a substantial amount of time on the revision. Regarding the current concerns, we have now performed

additional analysis to further solidify these findings. Detailed, point-by-point responses to this reviewer's concerns are below.

Specific concerns:

Fig 2

The authors still have not described what they mean by the “GSEA algorithm”. Are they referring to the actual GSEA method - Subramanian PNAS which would require computing an enrichment score by applying a modified KS statistic to a distribution of r^2 values for all genes (ie significant vs non-significant) and comparing this enrichment score to sample label permutations. If so, they should describe how they adapted this method to chip-seq analysis or supply a reference to a method that does.

My guess is that they associating the “top 500 peaks” with annotated MSigDb gene sets and getting gene set p values using a hypergeometric test. This is not GSEA but just a hypergeometric test and should be described as such in the methods. Also they should justify why they used “top 500” instead of a more standard threshold for choosing top peaks / genes (i.e. FDR, q value, bonferroni corrected P) as inputs to this kind of analysis.

We thank the reviewer for pointing this out and apologize for the confusion around the pathway enrichment analysis method we used in this manuscript. The reviewer is correct. In the previous version, to calculate the enrichment P-values, we performed a hypergeometric test to assess the significance of overlap between genes associated with a specific peak set (e.g. sensitive width, robust area, etc) and a pathway in the MSigDB database.

We understand that it is not the actual GSEA algorithm developed in Subramanian et al, PNAS 2005 and after carefully re-visiting the classic GSEA algorithm, we agree that it is good approach to evaluate the enrichment of pathways in genes associated with H3K4me3 dynamics. In addition, since the GSEA algorithm relies on a ranked list of genes based on their properties (e.g. associated with sensitive versus robust H3K4me3 peak width) instead of a specific subset of peaks (e.g. top 500 peaks with sensitive H3K4me3) determined by a threshold, it also resolves the issue with the usage of top 500 peaks in the pathway enrichment analysis. Thus, in the revised manuscript, we have redone all pathway enrichment analysis with the GSEA algorithm, replaced the related figures with results from the new analysis (Figure 2D-2J), and made corresponding textual changes. Notably, although a completely different algorithm is used, the conclusion on enrichment of cell type specific functions in sensitive width peaks in human cancer cells and robust width peaks in mouse liver remains unchanged (Response Figure 1). We have also included descriptions of the technical details in the Methods.

The analysis in Fig S7 should be in the main text, to replace Figs 2D-E. From a hypothesis testing perspective, there are no differences between a set of significant hypotheses that have a median p value $1e-20$ vs a median p value / q value $1e-20$. The question you want to ask is whether “more gene sets are significantly perturbed” - if everything is significant it doesn’t matter if one condition has q values $1e-20$. They are essentially identical.

I think Figs S7C and S7D are the best way to address the hypothesis that sensitive width / robust width show the highest degree of “biological perturbation”. The results (that 70% of sensitive width sets and 40% of robust width sets relative to baseline of 0-10%) are more compelling than quantitative differences in p or q values. These should be shown in the main text, but with p values comparing pairs of bars in this plot (eg via fisher test or binomial test)). i.e. what matters more than a quantitative difference in Q values is that a significantly larger proportion of gene sets are significant (subject to a fixed, reasonable p / q value threshold) among the peak types of interest.

We thank the reviewer for raising this concern. As we noted in the response to the previous comment, we have now used pathway enrichment analysis with the GSEA algorithm instead of the hypergeometric test that we used in the previous version. In the GSEA algorithm, only genes included in the reference set (i.e. all genes with H3K4me3) are considered thus being able to correct for the background.

We completely agree with the reviewer that the previous figures S7C and S7D are the best way to demonstrate that sensitive width in human cancer cells and robust width in mouse liver have the strongest association with biological functions. Thus, we have done similar analysis with the GSEA algorithm, in which we counted the number of significantly enriched (GSEA FDR Q-value <1e-5) pathways in peaks with each category of dynamics (Response Figure 1A, 1D). The new results show that sensitive width in human cancer cells and robust width in mouse liver still account for most enriched pathways. We have included these results in the revised manuscript (Figure 2).

Panels B and C -

The use of bar charts is not clear here, especially to indicate extreme p values. The number of peaks that are included in such an analysis will make the weakest correlation appear very significant, and indeed here the correlation is not very strong ($R^2 = 0.09$). In addition, the reader is tempted to draw comparisons between pairs of bars, but no hypothesis test is being communicated. Presumably, the authors want to show that there is a significant but weak correlation between expression and H3K4me3 that is not sensitive to MR. (ie the correlation appears similar between red and blue bars in both cancer cell line and mouse liver). One hypothesis test would be to test for differential correlation between H3K4me3 peak size descriptors and gene expression across the two conditions, e.g. using a linear model or ANOVA with gene expression as a dependent variable, height and/or area and/or width as independent numeric variables and MR as a binary factor.

We thank the reviewer for raising this point. The main conclusion that we can draw from Figure 4B and 4C is that gene expression levels significantly correlate with all three H3K4me3 peak descriptors under both control and MR conditions, which is supported by positive correlation coefficients and the magnitudes of $-\log_{10}(P\text{-values})$ in the bar plots. We don't expect a very strong correlation between H3K4me3 and gene expression because gene expression is known to be affected by other factors including binding of transcription factors and existence of other epigenetic marks. It is also not necessary for the exact values of correlation coefficients to be identical in the two conditions.

We agree with the reviewer that a more rigorous statistical hypothesis test will provide additional information on whether the exact values of correlation coefficients between gene expression and H3K4me3 are robust to MR. Thus, we performed an additional random permutation test and computed p-values to assess the differential correlation between control and MR conditions (Response Figure 2). We note that although 5 out of the 6 p-values were below the significance level 0.05, they only imply that the differences of the correlation coefficients between control and MR conditions were significant. They are still consistent with

the conclusion that the weak yet significant correlations exist between gene expression levels and all three peak descriptors under both control and MR conditions in both models.

More importantly, the influence of peak width on gene expression seems to be the weakest of all three size descriptors. How does that play with the story developed by the authors - i.e. that peak

width encodes important aspects of H3K4me3 biology? This important inconsistency does not seem to be addressed by the authors.

We thank the reviewer for raising this concern and apologize for the confusion. We agree that the correlation between peak width and gene expression in the same condition is the weakest among the three size descriptors, which is also consistent with the well-known association between H3K4me3 enrichment (corresponding to peak height) and active transcription (Nature 2002, PMID: 12353038). Nevertheless, regarding the changes between control and MR conditions, the strongest signal is in peak width. This is the main finding of this study, which is supported by the enrichment of cell type specific biological pathways and TF binding sites in peak subsets with sensitive width in human cancer cells and robust width in mouse liver. Thus, the finding that peak width change is the only predictor of gene expression change under methionine restriction is consistent with the conclusions.

Panels D and E - It's not clear what the biological significance of steady state MR gene expression as a function of peak category. For sensitive width genes in human cancer, it would seem to make more sense to examine differential expression in MR vs normal relative, i.e. jump to 4J and/or even demonstrate a scatter or density plot of peak width vs gene expression showing the correlation. As for robust width genes - what does it mean that they are the most highly expressed in MR Mouse Liver (among the peak categories investigated). Are these also highly expressed in normal mouse liver? If these robust width genes are not differentially expressed, then are they truly biologically significant or important in MR? Or is the complement of this set important (non robust width, which are presumably differentially expressed).

We thank the reviewer for this comment. We believe that the significantly higher expression levels in genes associated with H3K4me3 width changes suggest that these genes are important for cancer cells or liver to maintain their cell type specific function, thus providing an additional layer of evidence supporting that changes in peak width encode information of cell identity.

For robust width genes in mouse liver, these genes were highly expressed in both control and MR conditions although some of them were differentially expressed. In addition, consistent with the correlation between changes in H3K4me3 peak width and changes in gene expression, these genes were more likely to be up-regulated and less likely to be down-regulated under MR. Similarly, genes with sensitive width in human cancer cells were more likely to be down-regulated (Response Figure 3).

To show that down-regulated genes (associated with sensitive width) in human cancer cells and up-regulated genes (associated with robust width) in mouse liver are more biologically significant, we have performed GSEA analysis in up-regulated and down-regulated genes in both human cancer cells and mouse liver. We have found that, compared to up-regulated genes in human cancer cells and down-regulated genes in mouse liver, down-regulated genes in human cancer cells and up-regulated genes in mouse liver were enriched with more biological pathways among which a substantial fraction was cell type specific (Response Figure 3). We have included these results in the revised manuscript (Supplementary Figures 11G-11L).

Panels J and K - The authors should not be bar plotting $-\log_{10} p$ values, since $-\log_{10} P$ magnitude differences have little meaning in this range. Moreover the plotting is visually confusing as one is trying to compare different bars - concealing the key patterns in the data. Furthermore, the authors should (like in 4B and 4C) perform hypothesis testing to ask which of these factors (height, area, width) is best predictive of gene expression (e.g. using a linear model) and providing p values on pairs of bars. How do those predictions play with their hypotheses from Figs 1-3. It seem, for example, in Panel J that Width would be best predictive of MR dependent gene expression changes in cancer cell lines, consistent with the authors assertion that peak width changes are biologically important here.

We thank the reviewer for the helpful suggestions. We completely agree with the reviewer that the magnitude of p-values here is mostly a consequence of a very large sample size and a more

A			B		
Human Cancer Cells			Mouse Liver		
Variable	Linear Coefficient	P-value	Variable	Linear Coefficient	P-value
Height Change	-0.092	0.16	Height Change	-0.058	0.084
Area Change	0.085	0.17	Area Change	0.62	6.79e-78*
Width Change	0.68	3.75e-15*	Width Change	0.24	8.39e-7*

Response Figure 4 (A)
Linear coefficients and P-values in the linear regression model to predict changes in gene expression from changes in H3K4me3 peak height, area, width in human cancer cells. (B) Same as in (A) but for mouse liver.

rigorous statistical hypothesis test will better elucidate the importance of peak width changes. Following the reviewer's suggestion, we have now evaluated the predictability of changes in gene expression from changes in peak height, area and width in both human cancer cells and mouse liver using a linear model and now show the p-values assessing significance of non-zero linear coefficients (Response Figure 4). The linear model shows that width change is the only variable with significant non-zero linear coefficients to predict changes in gene expression in both human cancer cells and mouse liver. We have included these new results in the revised manuscript (Supplementary Figure 11C, 11D).

Panels K - If mouse liver gene peak width changes significantly correlate with gene expression does that mean that genes in robust width peaks are not differentially expressed in MR? How do the authors reconcile this with their model.

We thank the reviewer for this comment. The significant correlation between changes in gene expression and changes in peak width indicates that genes with robust peak width are likely to be up-regulated and unlikely to be down-regulated. We believe that it reflects the robustness of normal organ functions in response to alterations in environmental factors. In other words, genes participating in maintaining normal functions of healthy liver tend to keep sufficient expression levels under changes of environmental variables such as dietary factors.

As we noted in the response to the previous comment raised by this reviewer (Response Figure 3), we have now compared the fractions of down-regulated and up-regulated genes in genes with robust width to the corresponding fractions in other genes with H3K4me3 and found a significantly smaller fraction of down-regulated genes and a significantly larger fraction of up-regulated genes in genes with robust width in mouse liver. Furthermore, the up-regulated genes were significantly enriched with more pathways which were also liver specific. Taken together, these results suggest that genes with robust peaks tended to be up-regulated under MR and were included in maintaining liver-specific functions. We have added these new results and relevant discussion to the revised manuscript.

Small critiques

- Line 161 "for each subset of peaks the distribution of the enrichment Q values of the top 100 enriched (one-sided fisher's exact Q values < 0.05) pathways"

This does not make sense. Which of the two is it? The top 100 pathways or all pathways with Q < 0.05? If top 100 then why 100? A Q value threshold is more rigorous / standard. If both, then something is wrong in the analysis - since Q<0.05 should give different numbers of pathways in each peak subset.

We thank the reviewer for raising this concern. The reason that we used the top 100 enriched pathways in the previous version of manuscript is because 100 is the maximal number of

significant pathways displayed in the output of the MSigDB website (<http://software.broadinstitute.org/gsea/msigdb/annotate.jsp>) for the pathway enrichment analysis. These pathways all have enrichment Q -values < 0.05 but there are other pathways also enriched although with less significance. We completely agree with the reviewer that a Q -value threshold is more rigorous in determining the set of enriched pathways. In the revised manuscript, we have replaced this part of analysis with pathway enrichment analysis using the GSEA algorithm, in which we also used GSEA FDR Q -value $< 1e-5$ as the threshold for significantly enriched pathways instead of top 100 pathways with lowest Q -values.

- This awkward sentence does not make sense ..

> Lines 70-72

> We found that aspects of the peak geometry such as the height and area were overall reduced
> that accounted for most of the global changes.

We thank the reviewer for raising this comment. We have rephrased this sentence to improve its clarity in the revised manuscript.

"nearly all aspects of H3K4me3 biology" is too broad of a claim, please tone down or clarify ...
"all aspects of H3K4me3 biology" is a concept that is hard to qualify, but I imagine would include features that would not be captured by a ChIP-seq profiling study, for example kinetics, protein complex interactions, and three dimensional nuclear biology. This statement is even harder to justify when one considers some of the more perplexing conclusions of this work.

- Suggested replacement e.g. "reflected important cellular processes previously linked to H3K4me3 histone modifications"

> Lines 72-75

> "Strikingly, however, while the most conserved feature of H3K4me3 dynamics was found in the > peak width, changes in peak width but not other features of peak geometry were associated
> with nearly all aspects of H3K4me3 biology including cell identity related gene expression
> programs and the dynamics of gene expression."

We appreciate the reviewer for suggesting these clarifications. We have made changes to the text according to the reviewer's suggestions.

REVIEWERS' COMMENTS:

Reviewer #4 (Remarks to the Author):

The authors have adequately addressed my concerns.